



# Total water vapour columns derived from Sentinel 5p using the AMC-DOAS method

Tobias Küchler[1], Stefan Noël[1], Heinrich Bovensmann[1], John P. Burrows[1], Thomas Wagner[2], Christian Borger[2], Tobias Borsdorff[3], and Andreas Schneider[3,a]

[1]University of Bremen, Bremen, Germany
[2]Max Planck Institute for Chemistry, Mainz, Germany
[3]SRON Netherlands Institute for Space Research, Utrecht, the Netherlands
[a]now at Earth Observation Research Unit, Finnish Meteorological Institute, Sodankylä, Finland

**Correspondence:** Tobias Küchler (kuechler@iup.physik.uni-bremen.de)

**Abstract.**

Water vapour is the most abundant natural greenhouse gas in the Earth's atmosphere and global data sets are required for meteorological applications and climate research. The Tropospheric Ozone Monitoring Instrument (TROPOMI) onboard Sentinel 5 Precursor (S5P) launched on 13 October 2017 has a very high spatial resolution of around 5 km and a daily global

coverage. Currently, there is no operational total water vapour product for S5P measurements. Here, we present first results of a new scientific total column water vapour (TCWV) product for S5P using the so-called Air Mass Corrected Differential Optical Absorption Spectroscopy (AMC-DOAS) scheme. This method analyses spectral data between 688 and 700 nm and has already been successfully applied to measurements from the Global Monitoring Experiment (GOME) on ERS-2, the Scanning Imaging Absorption Spectrometer for Atmospheric Chartography (SCIAMACHY) on Envisat and GOME-2 on MetOp.

The adaptation of the AMC-DOAS method to S5P data especially includes an additional post-processing procedure to correct the influences of surface albedo, cloud height and cloud fraction. The quality of the new S5P AMC-DOAS water vapour product is assessed by comparisons with data from GOME-2 on MetOp-B retrieved also with the AMC-DOAS algorithm and with four completely independent data sets, namely re-analysis data from the European Centre for Medium range Weather Forecast (ECMWF ERA5), data obtained by the Special Sensor Microwave Imager and Sounder (SSMIS) flown on the Defense

Meteorological Satellite Program (DMSP) platform 16 and two scientific S5P TCWV products derived from TROPOMI measurements. Both are recently published TCWV products for S5P provided by the Max Planck Institute for Chemistry (MPIC) in Mainz and the Netherlands Institute for Space Research (SRON), Utrecht. The SRON TCWV is limited to clear sky scenes over land.

These comparisons reveal a good agreement between the various data sets but also some systematic deviations between all

of them. On average, the derived offset between AMC-DOAS S5P TCWV and AMC-DOAS GOME-2B TCWV is negative (around -1.5 kg m$^{-2}$) over land and positive over ocean surfaces (more than 1.5 kg m$^{-2}$). In contrast, SSMIS TCWV is on average lower than AMC-DOAS S5P TCWV by about 3 kg m$^{-2}$.

TCWV from ERA5 and S5P AMC-DOAS TCWV comparison shows spatial differences over both land and water surface. Over land there are systematical spatial structures with enhanced discrepancies between S5P AMC-DOAS TCWV and ERA5



TCWV in tropical regions. Over sea, S5P AMC-DOAS TCWV is slightly lower than ERA5 TCWV by around $2\,\mathrm{kg\,m^{-2}}$. The S5P AMC-DOAS TCWV and S5P TCWV from MPIC agree on average within $1\,\mathrm{kg\,m^{-2}}$ over both land and ocean. TCWV from SRON shows differences to AMC-DOAS S5P TCWV of around $1.2\,\mathrm{kg\,m^{-2}}$. All of these deviations are in line with the accuracy of these products and with the typical range of deviations of $5\,\mathrm{kg\,m^{-2}}$ obtained when comparing different TCWV

data sets.

The AMC-DOAS TCWV product for S5P provides therefore a valuable new and independent data set for atmospheric applications which also shows a better spatial coverage than the other S5P TCWV products.

## 1 Introduction

As the most abundant natural greenhouse gas, water vapour has a strong impact on the energy balance of the atmosphere. Its

absorption of the upwelling thermal infrared radiation from the earth and the incoming solar radiation warms the atmosphere. Water vapour has a twice as strong greenhouse heating effect as carbon dioxide (Mitchell, 1989; Kiehl and Trenberth, 1997). Water vapour evaporates from the ocean and fresh water and also from vegetation and moist soil. When it condenses in the atmosphere to form clouds it releases latent heat.

The amount of water vapour in the atmosphere is limited by the saturated vapour pressure which depends on the temperature.

Thus changes in temperature will result in an altered water vapour loading. An increasing atmospheric temperature leads to an increase in water vapour saturation pressure which is given by Clausius-Clapeyron equation. In a warming climate there is more evaporation and thus the water vapour content in the atmosphere increases. This leads to a stronger absorption of outgoing long wave radiation, emitted from the earth's surface, and to an increase of temperature in the atmosphere. However, the scattering of clouds of the incoming solar electromagnetic radiation cools the surface (Boucher et al., 2013). Overall this

feedback mechanism is complex. Enhanced water vapour amounts will also affect the amount and strength of precipitation. As a consequence, the strength or amplitude of the hydrological cycle is also affected (Allan et al., 2014) Water vapour also plays an important role in atmospheric chemistry. In the atmosphere it is a source of the most important oxidizing agent, the free radical hydroxyl, OH.

In summary to understand the physics and chemistry of the atmosphere, the changing hydrological cycle and climate, it is

essential to know the global distribution of water vapour and its changes with time.

One of the most accurate methods to determine water vapour concentrations are in-situ measurements from radiosondes, which provide atmospheric profiles of various atmospheric constituents at selected locations. These sites are distributed globally, but most of them are on land. However, radiosondes measure local conditions and any network of such sondes is intrinsically sparse. The latter cannot fully capture the high spatial and temporal variability of water vapour from the local to the

global scale.

Total column water vapour (TCWV) is also retrieved using the Global Positioning System (GPS) satellite signals in combination with local GPS ground stations (Bevis et al., 1992; Rocken et al., 1993, 1995). One advantage is the temporal high





resolution. They yield TCWV for all weather conditions. In contrast, the spatial coverage is quite poor due to the limited amount of ground based receivers.

Another important part of the global observing system for water vapour are measurements made from passive remote sounding sensors from polar and geostationary orbiting platforms. These potentially provide global information about the atmosphere having full global coverage every day or better, dependent on the number of platforms flying simultaneously. This information can be used to fill the spatial and temporal gaps of the different ground based measurements. A variety of possible methods to derive the total water vapour amount from space has been developed for various spectral regions.

One of the earliest TCWV data sets provided by satellites was derived from measurements in the microwave region by Nimbus 5 on NOAA (e. g. Staelin et al., 1976). In the same spectral region the SSM/I instrument and its successor SSMIS on different platforms provide the longest TCWV times series from 1987 up to now. The measurements of microwave sounders yield water vapour under cloud free and cloudy conditions. These data products are usually limited to those measurements made above water surface. This is a result of the poor understanding of land surface emission in the microwave region. With microwave sounders it is possible to retrieve water vapour under cloud free and cloudy conditions, but the retrievals are usually restricted to water surfaces due to not well known contributions of land surface emissions to the received signal (Schlüssel and Emery, 1990; Wentz, 1997). However, Melsheimer and Heygster (2008) extended the microwave retrieval to polar regions where ice and snow is present throughout the year.

TCWV retrievals are also possible in the thermal infrared spectral region, e.g. by the mathematical inversion of measurements from Infrared Atmospheric Sounding Interferometer (IASI) (Schlüssel and Goldberg, 2002) or Landsat 8 (Ren et al., 2015).

In the near infrared retrievals are performed at wavelengths around 900 nm e.g. by the Medium Resolution Imaging Spectrometer (MERIS) (Bennartz and Fischer, 2001; Lindstrot et al., 2012) and its successor, the Ocean Land Color Instrument (OLCI) flown on Sentinel-3 (Preusker et al., 2021), or the Moderate Resolution Imaging Spectrometer (MODIS) (Sobrino et al., 2003; Diedrich et al., 2015). These methods are usually limited to highly reflective surfaces such as land, which excludes ocean areas with exception of sun glint cases.

Another alternative is to employ measurements made in the visible spectral range to compute TCWV from satellites. Noël et al. (1999) introduced a modified DOAS (Differential Optical Absorption Spectroscopy) approach applied to GOME measurements. This approach was also used to retrieve TCWV from the Scanning Imaging Absorption spectroMeter for Atmospheric CHartographY (SCIAMACHY) (Noël et al., 2005a, b) as well as from GOME-2 (Noël et al., 2008) on the MetOp series. Wagner et al. (2003) described another approach to retrieve TCWV using the DOAS technique for GOME in the visible red spectrum. Later, Wagner et al. (2013) described an approach to derive TCWV from GOME-2 and Ozone Monitoring Instrument (OMI) using the spectra from 430 to 450 nm. Advantageous for this method is a more homogeneous and higher surface albedo especially over water. As a consequence, the backscattered signal is stronger but the absorption strength of $H_2O$ is generally weaker. Wang et al. (2014) used a similar approach to determine TCWV from the Ozone Monitoring Instrument (OMI). They used a wider spectral range from 430 nm to 480 nm to include water vapour absorption at 470 nm.

In autumn 2017 the Sentinel-5 Precursor (S5P) satellite was launched. It contains the Tropospheric Ozone Monitoring Instrument (TROPOMI), which provides an unprecedented high spatial resolution and temporal sampling.





**Table 1.** Overview of the satellite TCWV data sets used in the study. CF is the cloud fraction, AMF is the air mass factor, AMCF is the air mass correction factor, SZA is the solar zenith angle.

| Dataset | GOME-2 on Metop-B | S5P | | SSMIS on DMSP F16 |
|---|---|---|---|---|
| Method | AMC-DOAS | MPIC | SRON | Wentz |
| Reference | Noël et al. (2008) | Borger et al. (2020) | Schneider et al. (2020) | Wentz et al. (2012) |
| Fit window | 688–700 nm | 430–450 nm | 2.354–2.38 μm | 3 channels* |
| Filter criteria | AMCF >= 0.8 | snow/ice filter | aerosol filter | rain filter |
| | SZA<=88.0° | AMF >= 0.1 | SZA <= 75° | |
| | | CF <= 0.2 | CF <= 0.01 | |
| Availability | Global | Global | Land | Sea |

*These channels are 19.35 GHz, 22.235 GHz and 37.0 GHz.

Currently, no operational S5P total column water vapour product exists. Schneider et al. (2020) presented a method to derive water vapour isotopes HDO and $H_2O$ from S5P data in the short-wave infrared (SWIR). Most recently, Borger et al. (2020) retrieved TCWV from Sentinel-5P in the blue spectral range. This is similar to the approach described by Wagner et al. (2013).

Fortunately, the spectral range around 700 nm, which is used in the AMC-DOAS retrieval, is also present in S5P spectra.
Therefore, it is also possible to apply the AMC-DOAS method to TROPOMI data and thus extend the existing time series of AMC-DOAS TCWV.

In the current paper we present first results from the adaptation of the AMC-DOAS algorithm to this new instrument. The paper is structured as follows: Section 2 gives an overview of used instruments and data. Section 3 entirely explains the adaption of AMC-DOAS to S5P measurements. In particular, its dependence on albedo and cloud properties will be evaluated
and corrected. In section 4 the results of the retrieval and comparisons to other data sets are presented. Section 5 gives the summary and the conclusion.

## 2 Data

This section describes all external data sets used in this study either for generation of the new S5P AMC-DOAS data product (see section 3) or for the comparisons with other TCWV data (see section 4).

### 2.1 Sentinel 5P Level 1 data

Sentinel-5P (S5P) is part of the European Commission's Copernicus programme and was launched on 13th October 2017. It is a low polar orbiting satellite observing Earth's surface and atmosphere at roughly 824 km height. The satellite crosses the equator at 13:30 local time in an ascending node.





TROPOMI onboard S5P is a nadir viewing spectrometer, which has a wide spectral range covering the ultraviolet (UV) and visible spectral range (270 nm to 500 nm), the visible / near infrared (NIR) from 675 nm to 775 nm and the shortwave infrared (SWIR) region from 2305 nm to 2385 nm (Veefkind et al., 2012). For most of the spectral channels the spectral resolution is about 0.5 nm with a sampling of around 0.1 nm. The first UV and the SWIR band have spectral resolutions of 1.0 nm and 5  0.25 nm, respectively.

The visible / near infrared bands are suitable for the retrieval of TCWV from S5P with the AMC-DOAS algorithm. In particular, radiances from Band 5 ranging from 661 to 725 nm are used in the present study. They are processed with the L0-1b data processor version 01.00.00. Irradiance data are taken from the corresponding S5P L1B data set closest in time before the radiance measurement.

10  S5P's swath width of 2600 km allows an almost full daily coverage even in tropical regions. Currently, the spatial resolution of the sensor is $5.5 \times 3.5 \, \text{km}^2$ except for SWIR bands ($5.5 \times 7.0 \, \text{km}^2$) such that in contrast to other satellite instruments mentioned in section 2.4 below finer features in TCWV are resolved.

After the launch of S5P on the 13 October 2017 up to the end of April 2018 all its sensors were tested and calibrated. During this commission phase data sets are not provided regularly. However, after switching to operational mode the delivery of the 15  radiances is almost continuous.

For the comparison studies more than two years of daily data is used. The time span of these data is from May 2018 to December 2020.

## 2.2 GMTED2010

The U.S. Geological Survey provides the Global Multi-resolution Terrain Elevation Data 2010 (GMTED2010) (Danielson and 20  Gesch, 2011) which is used to get information on surface height and its type on very fine resolution up to 7.5 arc-seconds. The data set used in this study is provided on a 0.025° times 0.025° spatial resolution and comprises surface type, surface elevation and surface roughness. For the AMC-DOAS product the closest match between the location of S5P measurement and the GMTED2010 data product is chosen. Surface type is used to distinguish between land and sea. The surface height is needed to derive the surface height dependent TCWV product.

25  ## 2.3 The S5P FRESCO product

The Fast Retrieval Scheme for Clouds from the Oxygen A band (FRESCO, Koelemeijer et al., 2001; Wang et al., 2008) is a method to derive cloud pressure or cloud height and cloud fraction. The method uses three different 1 nm wide spectral windows close to the oxygen A band near 760 nm with various absorption strength.

In the 758-759 nm window no oxygen absorption occurs. The measured signal thus depends mainly on the cloud albedo, 30  surface albedo and the cloud fraction. Within the $O_2A$ band at 760-761 nm with very strong oxygen absorption and at 765-766 nm with weaker oxygen absorption the reflected sunlight additionally depends on cloud top pressure. The depth of the $O_2$ A band gives an information of the height of the clouds. All three wavelength windows provide all necessary information to retrieve cloud height and cloud fraction.



In this study we use the cloud information from the operational FRESCO product for S5P (Apituley et al., 2017) for filtering and post-processing (see section 3). It is provided on version 1.002 to 1.04.

## 2.4 Water vapour data sets

The independent TCWV products used for comparison are briefly described in this section. An overview of the different
correlative satellite TCWV data sets used in this study is shown in Tab. 1.

### 2.4.1 GOME-2 AMC-DOAS TCWV

The first GOME-2 instrument on the MetOp series was launched on MetOp-A in October 2006 (Munro et al., 2016). It is an improved version of GOME on the second European Remote Sensing Satellite (ERS-2) (Burrows et al., 1999; Munro et al., 2006). GOME-2 observes the atmosphere in a spectral range from 240 nm to 790 nm with a spectral resolution of 0.26 nm to
0.51 nm. By default its spatial resolution is 80 km across track times 40 km along track with a swath width of 1920 km. Since the launch of MetOp-B in September 2012 both satellites fly in a tandem operation mode. The swath of GOME-2 on MetOp-A was then reduced to 960 km resulting in an increase of spatial resolution by a factor of two across track on the cost of spatial coverage. Metop-B has a sun-synchronous descending orbit at 9:30am local time of equator crossing. Since November 2018 MetOp-C completes the MetOp series.

AMC-DOAS water vapour products are available for all three MetOp sensors (see e.g. Noël et al., 2008), but for the comparisons with S5P data described in the current study the GOME-2 instrument on MetOp-B (version 0.5.5a) has been selected because it provides the best spatial and temporal coverage. The estimated accuracy of the GOME-2 TCWV depends on cloudiness and TCWV amount and is typically better than $5 \, \mathrm{kg \, m^{-2}}$.

### 2.4.2 SSMIS TCWV

From 1987 the Special Sensor Microwave Imager/Sounder (SSM/I) flew on satellites of the Defence Meteorological Satellite Program (DMSP). It measures radiances in discrete spectral bands at wavelengths near 1 cm. From 2003 onwards this series was succeeded by the Special Sensor Microwave Imager and Sounder (SSMIS) on various platforms up to F18 (Kunkee et al., 2008). For comparison studies with S5P presented in the current paper the dayside data from the SSMIS instrument on the DMSP F16 satellite are chosen. This is because it has a ascending orbit with an equator crossing time of 15:54 which fits best
to the S5P observation time.

Its swath width is around 1700 km. SSMIS total water vapour data used here are provided as daily gridded data (0.25° resolution) by Remote Sensing System (Wentz et al., 2012). SSMIS data are only available over water surface for rain free situations. The total water vapour product is processed with the algorithm of Wentz (1997) with version v7. The accuracy of the SSMIS TCWV is around $1 \, \mathrm{kg \, m^{-2}}$.





### 2.4.3 MPIC S5P TCWV

The Satellite Remote Sensing Group at the Max Planck Institute for Chemistry (MPIC) also provides a TCWV product from TROPOMI measurements making use of the water vapour absorption in the blue spectral range (Borger et al., 2020). The retrieval consists of the common two-step DOAS approach: in the first step the spectral analysis is performed for a fit window from 430-450 nm within a linearised scheme. Then, the retrieved slant column densities are converted to vertical columns using an iterative scheme for the water vapour a priori profile shape, which is based on an empirical parameterisation of the water vapour scale height. During an extensive theoretical error estimation, the retrieval's TCWV uncertainty has been approximated to about 10-20 % for favourable and 20-50 % for unfavourable observation conditions. Furthermore, in the framework of a validation study based on daily and hourly measurements it was demonstrated that the MPIC S5P TCWV product is in very good agreement to reference data sets (e.g. SSMIS) for clear sky scenarios over ocean as well as over land surface. For this study only measurements have been included for which the effective cloud fraction is between 0 and 0.2, the airmass factor > 0.1, and the snow-ice flag indicates snow- and ice-free conditions. The accuracy of the TCWV product is up to 25 % (2.8 kg m$^{-2}$) for TCWV smaller than 20 kg m$^{-2}$ and up to 15 % for TCWV larger than 20 kg m$^{-2}$.

### 2.4.4 SRON S5P TCWV

The Netherlands Institude for Space Research (SRON) provides a TCWV product that is restricted to clear sky scenes over land and separates water vapour isotops (H$_2$O/HDO) and is retrieved from the SWIR infrared measurements of TROPOMI from 2354 to 2380.5 nm (Schneider et al., 2020). More details about the retrieval approach and settings can be found in (e.g. Scheepmaker et al., 2016). The forward model used here ignores scattering which makes strict filtering of clouds necessary. As cloud filter data from the Visible Infrared Imaging Radiometer onboard Suomi National Polar-orbiting Partnership (Siddans, 2016) are used. The upper threshold for cloud cover is a cloud fraction of 1 %. An additional filter for aerosols is also applied. Values at solar zenith angles larger than 75° are discarded. The albedo of water surfaces is too low to retrieve TCWV over oceans such that TCWV only is used over land surfaces. In this study we use version 9_1 of this data set which shows a bias to TCCON stations of (0.06±0.9 kg m$^{-2}$ ((1.1±7.2) %).

### 2.4.5 ECMWF ERA5 TCWV

The ERA5 reanalysis data set (Hersbach et al., 2020) from the European Centre for Medium Range Forecast Reanalysis provides atmospheric parameters such as temperature and humidity computed on 137 levels from surface height to 80 km. It is a model data set in which a large variety of observational data including satellite measurements (e.g. SSMIS), radiosondes and ground stations are assimilated.

This product is available every hour; data used here are on a 0.25° spatial grid. TCVW is derived by vertical integration of the profile data.





## 3 Methods

### 3.1 AMC-DOAS Approach

The approach known as Differential Optical Absorption Spectroscopy was first used to describe active remote sensing measurements having long tropospheric optical paths. (Perner and Platt, 1979). Variants of DOAS techniques were proposed and have been successfully applied from space (see e.g. Burrows et al., 1999, and references therein) to derive the amount of trace gases in the atmosphere. The method uses the Lambert Beer's law, which describes the attenuation of light due to gas absorption along a light path. The amount of a trace gas along this light path is the slant column density. The slant column density is converted into a total vertical column via a so-called air mass factor. This air mass factor is usually derived from radiative transfer calculations taking the solar geometry and scattering processes in the atmosphere into account.

The standard DOAS approach is in principle only valid for weak absorbers. Water vapour is usually a strong absorber and has a highly structured absorption spectrum, which typically is not resolved by the measuring spectrometer. This causes saturation effects that have to be considered in the retrieval.

To account for this, Noël et al. (1999) developed a modified version of the standard DOAS method named Air Mass Corrected Differential Optical Absorption Spectroscopy (AMC-DOAS). This method uses the equation

$$\ln\left(\frac{I_\lambda}{I_{0,\lambda}}\right) = P - a \cdot \left(\tau_{O_2,\lambda} + c_\lambda \cdot C_v{}^{b_\lambda}\right) \tag{1}$$

where $I_{0,\lambda}$ and $I_\lambda$ are the solar irradiance and Earth's backscattered radiance, respectively. The index $\lambda$ denotes quantities with dependence on the wavelength. $\tau_{O_2,\lambda}$ is the optical depth of oxygen. The quantity $c_\lambda$ contains the absorption cross section and air mass factor. The exponent $b_\lambda$ describes saturation effects in the spectra. As in standard DOAS, $P$ is an low order polynomial accounting for broadband features like scattering. $\tau_{O_2,\lambda}$, $b_\lambda$ and $c_\lambda$ are spectral quantities which are pre-calculated using a radiative transfer model.

$a$ is the so-called air-mass correction factor, which accounts for differences between the real atmospheric conditions / light path compared to those assumed in the radiative transfer calculations. $C_v$ is the total vertical column of water vapour, which is derived together with $a$ and $P$ by a nonlinear fit.

This method is applied to the measured $I_\lambda$ and $I_{0,\lambda}$ at a spectral range of 688–700 nm, which has been selected, because in this spectral region absorption lines of oxygen and water vapour are both present and of similar strength. This is important, because the underlying assumption of AMC-DOAS is that the same correction factor $a$ can be applied to both oxygen and water vapour. This will be explained in more details in the following.

In the case of perfect match of model conditions to the true atmospheric conditions no correction needs to be made. In this case the measured optical depth of oxygen equals the modelled $\tau_{O_2}$ thus $a = 1.0$. If there is a deviation (e. g. introduced by different meteorological conditions) the light path and thus the oxygen absorption depth differs to the modelled one. Hence the correction factor $a$ differs from one. In this case it scales the spectra such that the modelled oxygen absorption and the measured one match. Because it is assumed that the effect of differences in light path is the same for water vapour and oxygen the same scaling factor can be applied to water vapour.





In currently existing applications of the AMC-DOAS method for the GOME-like instruments (Noël et al., 1999; Noël et al., 2005a; Noël et al., 2008) a surface elevation of 0 km and a constant surface albedo of 0.05 are assumed for the determination of the spectral parameters $\tau_{O_2,\lambda}$, $b_\lambda$ and $c_\lambda$ via radiative transfer calculations. The parameters are calculated for various solar zenith angles ranging from $0°$ to $88°$. During the retrieval, the quantities are then interpolated to the actual solar zenith angle

of the measurement. A fixed reference $H_2O$ profile for a tropical atmosphere with a TCWV of $41.8\,\mathrm{kg\,m^{-2}}$ from LOWTRAN (Anderson, 1995) is used. No clouds are included in the radiative transfer calculations, thus the retrieval is in general only valid for cloudfree scenes; however small amounts of clouds can in principle be handled via the air mass correction factor $a$.

The currently existing AMC-DOAS data sets for GOME, SCIAMACHY and GOME-2 use radiative transfer data bases derived from SCIATRAN version 2 (Rozanov et al., 2005) in combination with HITRAN 2004 spectral line data (Rothman

et al., 2005) are used. Modelled spectra are convoluted with a Gaussian slit function having an optimised full width at half maximum (FWHM) for each instrument (between 0.35 nm for GOME and 0.59 nm for GOME-2C) to account for the different spectral resolutions.

## 3.2   Adaption and optimization of AMC-DOAS to Sentinel-5p observations

For the application to S5P the AMC-DOAS method was adapted in the following way. The radiative transfer model SCIATRAN

v3.8 (Rozanov et al., 2014) in combination with the HITRAN 2012 (Rothman et al., 2013) spectral absorption database is used to compute the quantities $c$,$b$ and $\tau_{O_2}$. As the reference $H_2O$ vertical profile a tropical atmosphere with a TCWV of $41.8\,\mathrm{kg\,m^{-2}}$ is used (from LOWTRAN data base). The spectra are then convoluted with the ground pixel dependent instrument spectral response functions (ISRFs) (van Hees et al., 2018) of S5P. Their full width half maximum varies around and in the range given by $0.34\,\mathrm{nm} \pm 0.002\,\mathrm{nm}$. The spectral quantities are calculated for a reference surface albedo of 0.02 for an albedo of a water

surface. The surface height is also considered. As surface height reference the Global multi-resolution terrain elevation data 2010 (GMTED2010; Danielson and Gesch, 2011) is used. The radiative transfer database is then calculated for every ground pixel and various surface heights from 0 to 9 km. Note that this added dependence on the surface height also changes the definition of the AMC-DOAS water vapour product: The S5P TCWV is defined as the total column above the surface, whereas in previous AMC-DOAS products it was defined as the total column above sea level. This has the advantage that TCWV over

mountain ranges are valid data points.

In previous applications for the GOME-like instruments the scaling factor $a$ was also used as an inherent quality check (Noël et al., 1999). If the correction is too large (which is mainly due to clouds) the retrieval results are discarded. The corresponding minimum air mass correction factor of 0.8 is also used as filter criterium for the S5P data. However, it turns out that for S5P this filter is not effective enough; too many (especially cloudy) data remain. In general, we derive typically higher air

mass correction factors for S5P than for the other instruments. We attribute this mainly to the different equator crossing times (morning vs. noon) in combination with the higher spatial resolution and wider swath width of S5P. Thus additional filtering is needed.

The largest source of error in the AMC-DOAS TCWV product are associated with partially cloud filled ground scenes. The larger the fraction of cloud within a ground scene and the higher the cloud, then the lower the effective sampling of the



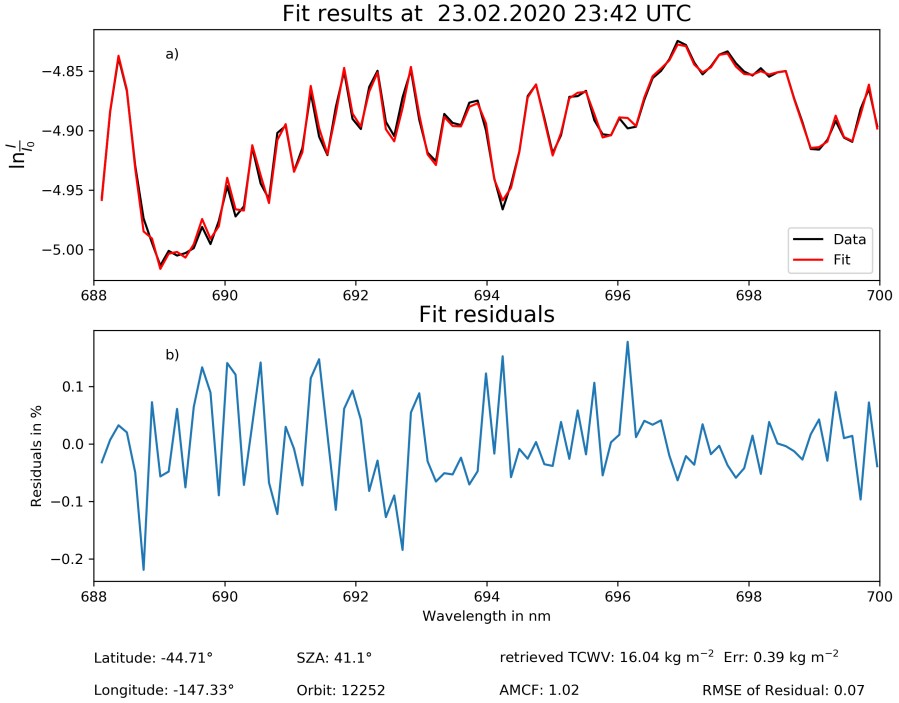

**Figure 1.** a) Example measurement (black) from S5P and fit (red). b) Relative fit residual (relative difference between measurement and fit) in percent.

troposphere. We therefore apply an additional cloud filter, which is based on cloud fraction and cloud height provided by the operational S5P FRESCO cloud product. A pixel is considered as cloudy if the cloud fraction is larger than 0.2. In addition, measurements with cloud heights above surface of more than 2.0 km are also discarded.

An example for a S5P measured spectrum and the corresponding fitted spectrum from the retrieval can be seen in Fig. 1a

5   for a scene over the pacific with very little cloud fraction. In this example the retrieved TCWV is $16.0 \, \mathrm{kg \, m^{-2}}$ with an retrieval error of $0.39 \, \mathrm{kg \, m^{-2}}$. The residual, which is given in relative amount (measurement minus fit divided by measurement, see Fig. 1b), is not larger than roughly $0.3 \, \%$ in this example. The root mean square of the absolute residual (measurement-fit) is with 0.07 very low. This shows that the measurement and the fit match very well.

### 3.3   Postprocessing

10   With the AMC-DOAS method one day of S5P measurements (23 February 2020) has been processed and filtered according to the procedure described above. The resulting S5P TCWV product shown in Fig. 2a represents all expected spatial features. Within the Intertropical Convergence Zone (ITCZ) the values are largest. Towards polar regions the TCWV decreases.

Details on the quality of the AMC-DOAS S5P TCWV are revealed by the deviation to the collocated ERA5 TCWV (Fig. 2b) which shows several issues. On global average, there is only very little difference of $0.05 \, \mathrm{kg \, m^{-2}}$ between both data sets. Over





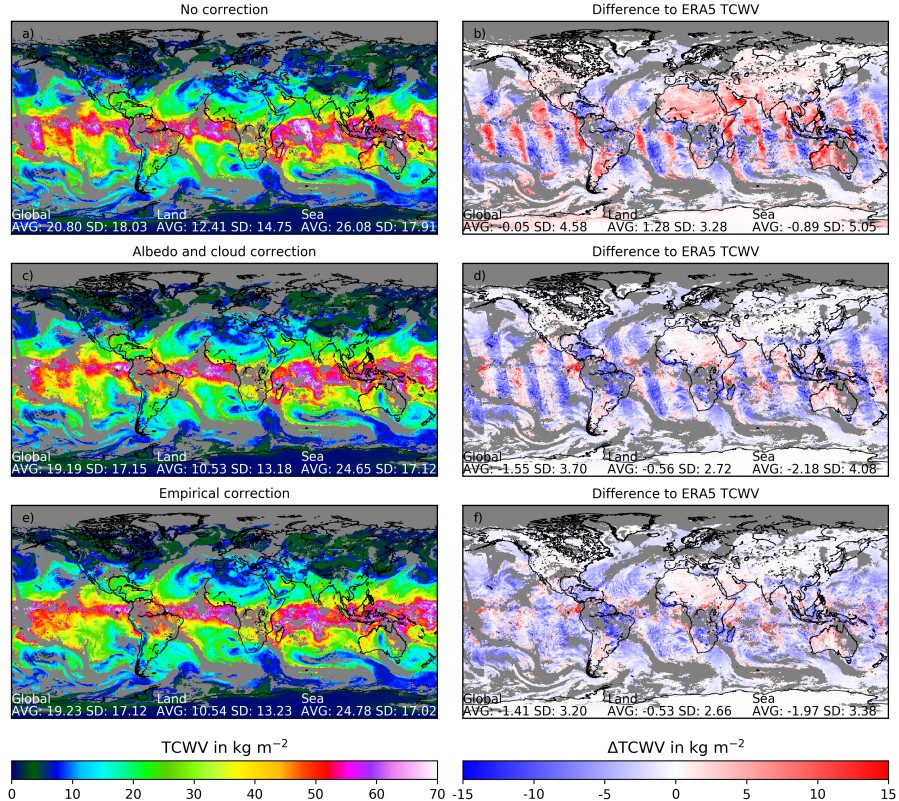

**Figure 2.** Visualisation of the effects of the correction on the absolute AMC-DOAS S5P TCWV in the left column and its deviation to TCWV from ERA5 in the right column for the 23 February 2020. Grey areas are data gaps mainly due to the filtering.

ocean repeating patterns are visible. These patterns are more pronounced over regions with higher TCWV. Over land systematic positive deviations over regions with higher surface albedo can be observed, like Sahara and Australia. These regions typically have a higher surface albedo than the reference used for the AMC-DOAS radiative transfer data base. This implies that surface albedo influences on the retrieved AMC-DOAS TCWV need to be considered. Also remnant clouds will affect the retrieval.

5    Thus an additional correction scheme has been introduced to reduce systematic effects due to surface albedo and clouds. This is described in the following subsections.

### 3.3.1 Albedo and cloud effects

Clouds hide parts of the atmospheric profile depending on cloud height and cloud fraction. This is especially critical for water vapour, which is most abundant close to the surface. This is the reason why we limit from present the scenes being studied to

10   those having cloud fractions from 0 to 0.2.

The effects of varying surface albedo on the measured signal and the light path are in principle handled in the AMC-DOAS method by the fitted polynomial and the air mass correction factor. The remaining influences of albedo on the retrieval results





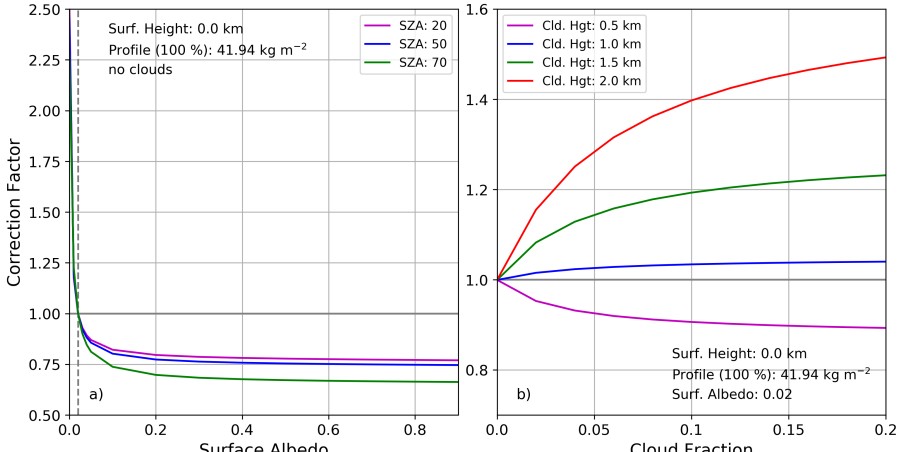

**Figure 3.** a) Correction factor as function of surface albedo for various solar zenith angles. b) Correction factor as function of cloud fraction for various cloud heights.

are due to the inequal (and usually unknown) shapes of the water vapour and oxygen profiles. The cloud effect and the albedo effect are not completely separable. It is therefore required to derive correction factors for various combinations of cloud fraction, cloud height, surface albedo, surface height and solar zenith angle.

To investigate the dependence of AMC-DOAS TCWV on surface albedo and cloud properties radiances $I_{clear}$ and $I_{cloud}$

are simulated with SCIATRAN for the clear sky case and the fully cloudy case, respectively. In this manuscript the term of albedo is used to describe the spectral reflectance from surface and clouds. This assumes a lambertian surface where the total reflected radiation is homogeneously distributed over a hemisphere, i. e. $2\pi$ steradians. For the small spectral window used in the retrieval, this is considered a reasonable approach and spectral dependence of surface albedo is ignored. For the clear sky case, surface albedo, surface height and solar zenith angle are varied. For the cloudy case we also consider dependencies on

cloud height and cloud fraction.

A cloud is considered in the simulations as a reflecting layer with an albedo of 0.8 which is located at a given height. This follows the definition of the S5P FRESCO product. The simulated cloud-free and cloudy radiances are then mixed by the cloud fraction $CF$ according to the independent pixel approximation:

$$I_{mixed} = CF \cdot I_{cloud} + (1 - CF) \cdot I_{clear} \qquad (2)$$

The spectrum $I_{mixed}$ is then used in the AMC-DOAS retrieval. The ratio of the reference ('true') TCWV $C_{v,ref}$ to the retrieved TCWV $C_{v,retr}$ may then be used as a multiplicative correction factor $c_{ac}$:

$$c_{ac} = \frac{C_{v,ref}}{C_{v,retr}} \qquad (3)$$

Examples for this correction factor are shown in Fig. 3.



If no cloud is present, only the variation of the surface albedo plays a significant role in the AMC-DOAS retrieval (Fig. 3a). In the case of smaller surface albedo than the reference of 0.02 the retrieved TCWV is underestimated thus the correction factor is larger than 1. Larger surface albedo is associated with an overestimation of the reference TCWV and a correction factor smaller than one.

An example for the correction factor for cloudy scenes is shown in Fig. 3b. For low level clouds located at around 500 m above surface or less there is an increase of the retrieved TCWV resulting in correction factors smaller than 1. Low level clouds only hide a relative small part of the water vapour profile. As discussed before albedo leads to an overestimation of total water vapour. Since the albedo of clouds is large the albedo effect overcompensates the shielding effect. This effect increases for larger cloud fractions. For clouds higher than 500 m the shielding effect dominates resulting in a reduction of the retrieved

TCWV and a correction factor larger than 1. It can also be seen that the correction increases for higher clouds and larger cloud fractions.

To avoid that the correction factor dominates the retrieval results an additional filter is applied to exclude situations where the correction is too large. Thus the overall correction factor is restricted to values between 0.6 and 1.2.

The final albedo and cloud correction factor depends on geometrical information (solar zenith angle, ground pixel; taken

from the S5P measurements), surface elevation (from GMTED2010), cloud fraction and cloud height (from the S5P FRESCO product) and surface albedo.

As the surface albedo is highly variable, we do not use a climatology but determine it directly from the S5P reflectance measurements from 684 nm to 686 nm. This spectral region is close to the retrieval window of 688-700 nm, but contains no major atmospheric trace gas absorption. To relate the reflectance to the surface albedo radiances and irradiances are simulated

from 684 nm to 686 nm with varying surface albedo, solar zenith angle,surface height, cloud fraction and cloud height. To smooth out fluctuations the average reflectance over this 2 nm window is calculated. This results in a database from which for each (measured) average reflectance, geometry and cloud properties a surface albedo can be derived via interpolation.

The resulting clear sky albedo and cloud correction is then applied as multiplicative factor ($C_{v,ac}$) to get the corrected TCWV:

$$C_{v,ac} = C_{v,uc} \, c_{ac} \tag{4}$$

where $C_{v,uc}$ is the uncorrected TCWV. Note that due to this correction the TCWV product is independent of the surface albedo chosen as reference for the basic AMC-DOAS retrieval.

The results of this correction when applied to the uncorrected data from 23 February 2020 is shown in Fig. 2c and its deviation to ERA5 in Fig. 2d. The global mean deviation is slightly increasing but the variability denoted by the standard

deviation SD is lower compared to the uncorrected product. Over land the application of the correction factors reduces the deviations over the deserts. However, over ocean there are still some patterns visible. These stripe-like deviations resemble orbital features; in the eastern part of the S5P swath the TCWV is generally lower than ERA5.





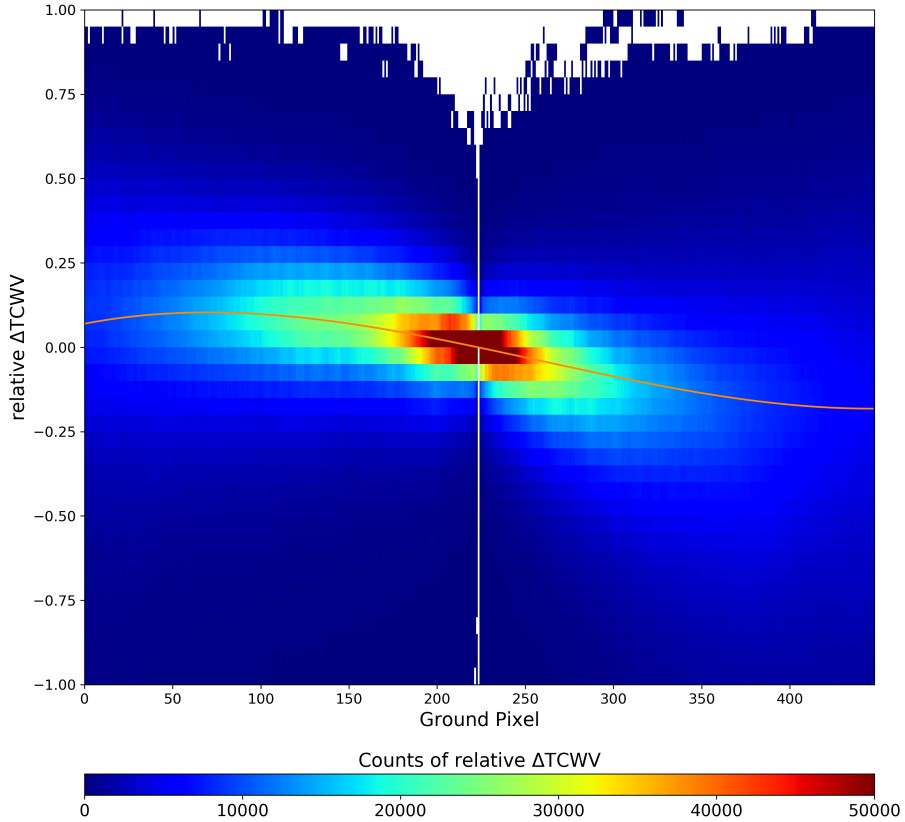

**Figure 4.** Counts of the relative deviation of the albedo and cloud corrected TCWV to the nadir value for every ground pixel for February 2020. The orange line is a fitted polynomial of 3rd degree

### 3.4 Empirical correction for S5P instrumental striping

The stripe-like deviations over ocean shown in Fig. 2d cannot be reproduced by our simulation. Consequently, we assume that they are related to instrumental features. To eliminate this repeating pattern an empirical correction is performed. This correction only depends on the relative location of the ground pixel. There is no dependence on season or position of the instrument. Since these features are only visible over ocean and not over land the correction will only be applied to ground pixels located over water surfaces.

For this purpose for each swath over water surface the relative difference $\Delta C_{v,ac}$ of the retrieved TCWV at each ground pixel $i$ to the nadir value (ground pixel $i_{nadir}$=223) is computed:

$$\Delta C_{v,ac}(i) = \frac{C_{v,ac}(i) - C_{v,ac}(i_{nadir})}{C_{v,ac}(i)} \tag{5}$$

All S5P orbits in February 2020 are used for this to have good statistics. For every ground pixel with valid TCWV measurement $\Delta C_{v,ac}(i)$ is calculated and counted with bins of 0.05. This results in a histogram of $\Delta C_{v,ac}$ as function of ground pixel





number which is shown in Fig. 4. As can be seen there is a systematic ground pixel dependence of $\Delta C_{v,ac}$. In the western part of the swath (ground pixel numbers smaller than $i_{nadir}$) the relative deviation is positive whereas the eastern part (ground pixel numbers larger than $i_{nadir}$) shows more pronounced negative deviations.

For the correction the maximum amount of $\Delta C_{v,ac}$ at each ground pixel is then used to fit a polynomial $P_{emp}$ of third degree
(orange line in Fig. 4):

$$P_{emp}(k) = a_0 + a_1\ k + a_2\ k^2 + a_3\ k^3 \tag{6}$$

where $k = i - i_{nadir}$ is the shifted ground pixel number and $a_j$ are the derived polynomial coefficients, namely: $a_0 = 0.0$, $a_1 = -1.099 \cdot 10^{-3}$, $a_2 = -1.13 \cdot 10^{-6}$ and $a_3 = 1.075 \cdot 10^{-8}$ .

The multiplicative correction factor $c_{emp}$ for every ground pixel is then defined as:

$$c_{emp}(i) = 1 - P(i - i_{nadir}) \tag{7}$$

This correction is then applied in addition to the cloud and albedo correction, leading to:

$$C_{v,emp} = C_{v,ac}\ c_{emp} \tag{8}$$

The results are shown in Fig. 2e and f. The spatial patterns over ocean are corrected out and also the mean deviation to ERA5 and the scatter of the data is reduced.

Note that all applied corrections do not significantly change the main vapour patterns (Fig. 2a,c,e), but generally result in a smoother spatial distribution.

## 4  Results and Discussion

All S5P radiances from May 2018 to December 2020 have processed by the AMC-DOAS method and corrected as described above. The typical precision of the AMC-DOAS S5P TCWV for a single measurements (derived from the fit) is
about $0.5\,\mathrm{kg\,m^{-2}}$.

From these, a daily gridded data product with a spatial resolution of $0.25°$ degree$\times\,0.25°$ is produced, resulting a data set called $\mathrm{TCWV_{AMC,S5P}}$ in the following. An overview of this TCWV product is given in Fig. 5 which shows the spatial distribution of $\mathrm{TCWV_{AMC,S5P}}$ for four months.

The general features shown in the maps meets the expectations from climatology. In the tropics there is higher TCWV due
to high temperature. Within the ITCZ the values are highest. Towards the polar regions the air gets colder thus the TCWV decreases. The propagation of the main features during the course of the time is also visible. The global average of TCWV is around $18.5\,\mathrm{kg\,m^{-2}}$.

In January, the ITCZ is located close to the equator. During the course of time it shifts northwards until July. Large changes are observed comparing January and July over southeast Asia (e.g. India, China) and nearby water surfaces. Here the ITCZ
reaches its northernmost position causing an increase of $\mathrm{TCWV_{AMC,S5P}}$ of around $30\,\mathrm{kg\,m^{-2}}$ from January to July.





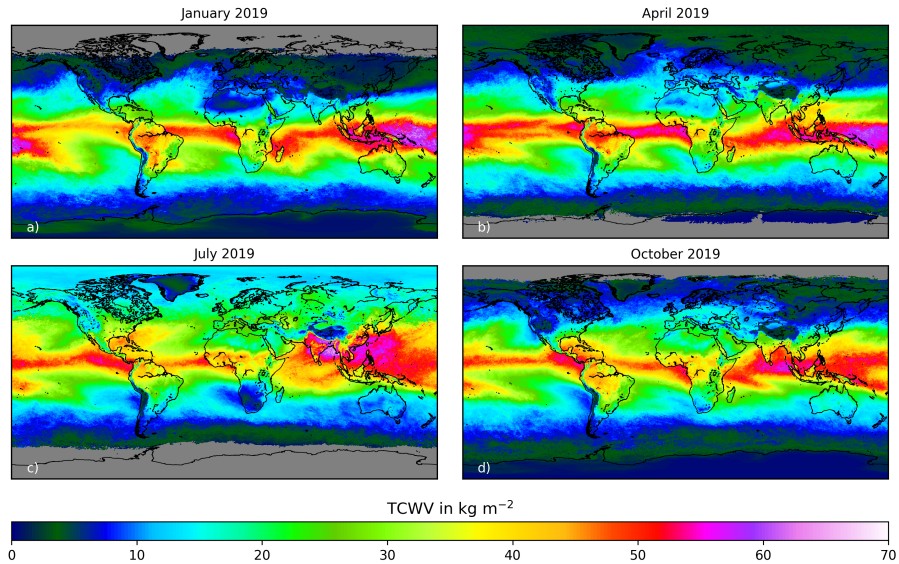

**Figure 5.** Global maps of mean TCWV$_{\text{AMC,S5P}}$ for four months in 2019.

During northern summer as the entire northern hemisphere warms the global average TCWV is largest ($23.1\,\text{kg}\,\text{m}^{-2}$). This is in large part due to larger land masses in the northern hemisphere. They are significantly warmer during July than the large oceans in the southern hemisphere in January. Smaller contributions come from Arctic regions which also show enhanced TCWV. Such large TCWV increase cannot be observed from July to January over the southern hemisphere due to the lack of 5 landmasses.

TCWV$_{\text{AMC,S5P}}$ also shows some differences between April and October. In general, all features follow the position of the sun but with a time lag of several weeks. That means in April the northern hemisphere is colder which results in higher TCWV in October.

Over sea the averaged TCWV$_{\text{AMC,S5P}}$ is higher than over land due to different surface elevation, larger temperature variability 10 and also large evaporation over water surfaces. No data are available in winter hemisphere's polar night region due to lack of solar insulation.

To assess the quality of this new data set it is compared to various other other data sets (see section 2.4 for more information), which are either also provided on a daily $0.25° \times 0.25°$ grid or have been gridded accordingly. We use the following notation for these correlative data sets:

15     – TCWV$_{\text{AMC,GOME-2B}}$:

        GOME-2B data product, which is based on the original AMC-DOAS approach; daily gridded to $0.25°$ degree$\times\,0.25°$.

    – TCWV$_{\text{WENTZ,SSMIS}}$:

        SSMIS data product using microwave emissions as input; provided on a daily $0.25°$ grid.





**Table 2.** Correlation coefficient $R$, slope $m$, intercept $n$, average difference $\Delta$TCWV (in $\mathrm{kg\,m^2}$) to the AMC-DOAS S5P product and collocation counts for the scatter plots in Fig. 6. The errors represent one standard deviation.

| Data set | Surface | R | m | n | $\Delta$TCWV | Counts |
|---|---|---|---|---|---|---|
| ERA5 | Land | 0.98 | $1.05\pm3.75\cdot10^{-4}$ | $0.00\pm5.76\cdot10^{-3}$ | -0.7±3.2 | 251160 |
| | Water | 0.98 | $0.96\pm3.16\cdot10^{-4}$ | $2.97\pm9.10\cdot10^{-3}$ | -2.0±3.5 | 354614 |
| AMC-DOAS GOME-2B | Land | 0.97 | $1.05\pm6.18\cdot10^{-4}$ | $0.46\pm7.91\cdot10^{-3}$ | -1.3±4.2 | 207626 |
| | Water | 0.92 | $0.93\pm7.99\cdot10^{-4}$ | $-0.26\pm2.00\cdot10^{-2}$ | 1.7±6.7 | 237105 |
| MPIC S5P | Land | 0.95 | $0.92\pm9.82\cdot10^{-4}$ | $0.75\pm2.27\cdot10^{-2}$ | 0.8±4.7 | 103960 |
| | Water | 0.97 | $0.93\pm3.77\cdot10^{-4}$ | $2.19\pm1.11\cdot10^{-2}$ | -0.3±4.1 | 329936 |
| SRON | Land | 0.99 | $0.89\pm4.02\cdot10^{-4}$ | $0.51\pm5.26\cdot10^{-3}$ | 0.8±1.4 | 52070 |
| SSMIS | Water | 0.96 | $0.99\pm5.43\cdot10^{-4}$ | $3.76\pm1.48\cdot10^{-2}$ | -3.7±4.6 | 256188 |

- $\text{TCWV}_{\text{MOD,ERA5}}$:

  ECMWF ERA5 model data, provided every hour on a 0.25° grid. Based on the position of every S5P pixel the spatial and temporal nearest ERA5 TCWV is chosen. This results in a pseudo swath data set consisting of ECMWF data at geolocations of S5P which is filtered according to the AMC-DOAS filter criteria. These pseudo swaths are then daily gridded to 0.25° × 0.25° resolution.

- $\text{TCWV}_{\text{MPIC,S5P}}$:

  S5P TCWV product data from MPIC in Mainz using the 'blue' spectral range; provided on a daily 0.25° grid.

- $\text{TCWV}_{\text{SRON,S5P}}$:

  S5P TCWV product data from SRON using the SWIR spectral range; daily gridded to 0.25° degree× 0.25°.

In the first step, daily $\text{TCWV}_{\text{AMC,S5P}}$ data are compared to other daily TCWV products. Global deviation maps are then presented and discussed. The comparisons are done over land and ocean separately to detect possible systematic features arising from surface type and/or elevation. It has to be noted that all satellites have different time of overpass thus diurnal changes in TCWV may affect the comparison results.

## 4.1 Daily comparisons

The comparison procedure for the daily data is as follows. For every day $\text{TCWV}_{\text{AMC,S5P}}$ and the other TCWV products are collocated. From the collocated data sets pairwise differences $\Delta TCWV_{AMC,S5P-Z}$ are calculated:

$$\Delta TCWV_{AMC,S5P-Z} = TCWV_{AMC,S5P} - TCWV_Z \tag{9}$$

Here, the index $Z$ denotes the specific data set to compare with. Additionally, the difference is averaged and its variability is given by standard deviation SD. The averages are calculated with weighting according to the latitude.



A linear regression model using least square technique is applied to $\text{TCWV}_{\text{AMC,S5P}}$ and $\text{TCWV}_{\text{Z}}$. This gives the correlation coefficient $R$ and the regression parameters $n$ and $m$ denoting the intercept and the slope, respectively.

A scatter plot for 23 February 2020 is shown in Fig. 6 for the various TCWV products. All statistical parameters are given in Tab. 2.

### 4.1.1 ERA5

ERA5 is a model comprimising satellite data (e.g. SSMIS), radiosondes and weather observations for reanalysis. That makes $\text{TCWV}_{\text{MOD,ERA5}}$ a very robust TCWV product. Due to the fact that ERA5 is an hourly data set temporal mismatch is restricted to less than half an hour.

The comparison between $\text{TCWV}_{\text{AMC,S5P}}$ and $\text{TCWV}_{\text{MOD,ERA5}}$ (Fig. 6a,b) shows a very small difference of -0.7 $\text{kg m}^{-2}$ (Fig. 6a) over land. The values are orientated along the 1:1 line which is also denoted the by small standard deviation of 3.2 $\text{kg m}^{-2}$. Over sea (Fig. 6b) the difference is larger (-2.0 $\text{kg m}^{-2}$) than over land but the standard deviation (3.5 $\text{kg m}^{-2}$) and also the correlation coefficient is very similar. The correlation coefficients are above 0.98 indicating a very good agreement between both data sets.

### 4.1.2 GOME-2B

The comparison between $\text{TCWV}_{\text{AMC,S5P}}$ and $\text{TCWV}_{\text{AMC,GOME-2B}}$ (Fig. 6c,d) shows very good agreement between both data sets irrespective of whether the retrieved TCWV is over land or water surfaces. This is demonstrated by the regression line (solid line in Fig. 6c,d) being very close to the 1:1 line (dotted) and a correlation coefficent above 0.9.

The average TCWV difference $\Delta TCWV_{AMC,S5P-AMC,GOME-2B}$ is -1.3±4.2 $\text{kg m}^{-2}$ over land (Fig. 6c), i. e. close to zero (see Tab. 2 for further details). Because the slope of the regression line is nearly one there is only little dependence of the difference on the magnitude of the TCWV. Due to the large number of lower TCWV around 10 $\text{kg m}^{-2}$ the linear regression model is more weighted to these values. Over ocean (Fig. 6d) there is more variability in the difference, which is also indicated by the standard deviation of 6.7 $\text{kg m}^{-2}$. The average deviation is 1.7 $\text{kg m}^{-2}$. There is a land sea bias of 3 $\text{kg m}^{-2}$ at this day.

Both products are processed with AMC-DOAS. In contrast to $\text{TCWV}_{\text{AMC,GOME-2B}}$ the surface height is considered during the retrieval of $\text{TCWV}_{\text{AMC,S5P}}$, which explains higher values of $\text{TCWV}_{\text{AMC,GOME-2B}}$ over land. The additional postprocessing only done for $\text{TCWV}_{\text{AMC,S5P}}$ also affects the results.

Other sources that influences the comparison results are the filters applied to the $\text{TCWV}_{\text{AMC,S5P}}$. As mentioned above $\text{TCWV}_{\text{AMC,S5P}}$ data are filtered with an additional cloud filter, which is not applied to GOME-2B data. The propagation of large scale cloud decks like in lows or the well known stratocumulus cloud region is not that fast within the time difference of MetOp-B and S5P overpasses of four hours (at equator). These cloud decks are therefore located at similar positions for both overpass times. Thus cloud masking applied to $\text{TCWV}_{\text{AMC,S5P}}$ will also filter clouds by some degree from $\text{TCWV}_{\text{AMC,GOME-2B}}$ (as we only consider grid points where both instruments have data).



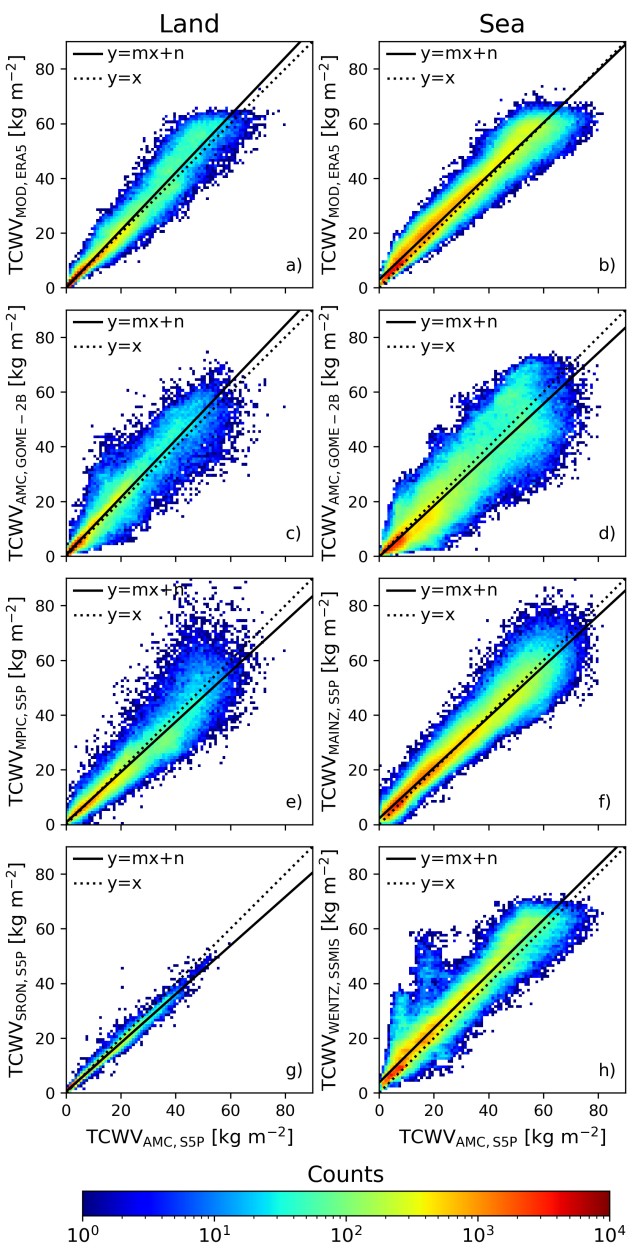

**Figure 6.** Density plot of TCWV comparisons with the AMC-DOAS S5P product over land (left column)) and over sea (right column) for 23 February 2020 for a) and b) ERA5 TCWV, c) and d) AMC-DOAS GOME-2B TCWV, e) and f) MPIC S5P TCWV, g) SRON S5P TCWV and h) SSMIS TCWV. The dotted line represents perfect agreement (1:1), the solid line shows the (fitted) linear relationship between the data sets. All statistical parameters are given in Tab. 2.



### 4.1.3 MPIC S5P

The $\text{TCWV}_{\text{AMC,S5P}}$ and $\text{TCWV}_{\text{MPIC,S5P}}$ both use S5P Level 1 measurements but different spectral regions are used in the retrieval. Over land (Fig. 6e) the mean difference between both data sets is $0.8\,\text{kg}\,\text{m}^{-2}$ with a standard deviation of $4.7\,\text{kg}\,\text{m}^{-2}$. The $\text{TCWV}_{\text{MPIC,S5P}}$ data contain a few values up to $90\,\text{kg}\,\text{m}^{-2}$ which are not observed in the $\text{TCWV}_{\text{AMC,S5P}}$. Over land there

are less valid data in $\text{TCWV}_{\text{MPIC,S5P}}$ compared to other data sets, which can be seen from the number of valid counts (Tab. 2) after collocation. This is due to the filtering of snow and ice contaminated scenes.

Over sea (Fig. 6f) there is almost no mean deviation ($-0.3\,\text{kg}\,\text{m}^{-2}$) and a lower variability ($4.1\,\text{kg}\,\text{m}^{-2}$) than over land. The differences are smallest over both land and sea compared to the other data sets. This meets the expectations because measurements are performed by the same instrument. Nevertheless, uncertainties may arise from sampling differences.

### 4.1.4 SRON S5P

The $\text{TCWV}_{\text{AMC,S5P}}$ and $\text{TCWV}_{\text{SRON,S5P}}$ also rely on the same instrument. In contrast to $\text{TCWV}_{\text{MPIC,S5P}}$ the SRON product has a poorer spatial coverage due strict cloud filtering and limitation to land. The main aim of the SRON data product is to provide columns with low error caused by cloud contamination. This results in 50000 collocated grid points over land (see Tab. 2) can be used for comparison. Fig. 6g also illustrates this. There is far less scatter visible than for the other data sets. Most of the

TCWV pairs are well oriented along the regression line. This is also shown by an almost perfect correlation coefficient of 0.99. On average, the deviation between $\text{TCWV}_{\text{AMC,S5P}}$ and $\text{TCWV}_{\text{SRON,S5P}}$ is $0.8\,\text{kg}\,\text{m}^{-2}$. The standard deviation is $1.4\,\text{kg}\,\text{m}^{-2}$ which is comparably low with respect to the other TCWV products; the $\text{TCWV}_{\text{SRON,S5P}}$ rarely exceeds $40\,\text{kg}\,\text{m}^{-2}$. Both low scatter and low total columns are probably related to the filtering, which removes almost all even partly cloudy scenes, i.e. especially those scenes which require dedicated corrections in the other S5P TCWV algorithms.

### 4.1.5 SSMIS

Microwave instruments are known to provide good information on total water vapour because microwave emission penetrates through clouds. However, the comparison to $\text{TCWV}_{\text{AMC,S5P}}$ is limited to ocean areas because SSMIS does not provided data over land surfaces.

The mean deviation between $\text{TCWV}_{\text{AMC,S5P}}$ and $\text{TCWV}_{\text{WENTZ,SSMIS}}$ is the largest compared to other sensors ($-3.7\,\text{kg}\,\text{m}^{-2}$).

The regression line in Fig. 6i shows a constant offset between both data sets. There are several possible reasons to explain this large offset:

 – The DMSP F16 has an orbit later in the afternoon (around 16:00 LT). This can affect the TCWV due to slight warming of the sea surface and the above air. This causes enhanced evaporation and thus a slightly higher water vapour content.

 – In the microwave region the radiation penetrates through the clouds. As consequence SSMIS senses the entire profile

also if clouds are present. The total water vapour usually is higher in cloudy scenes than in clear sky scenes which is also





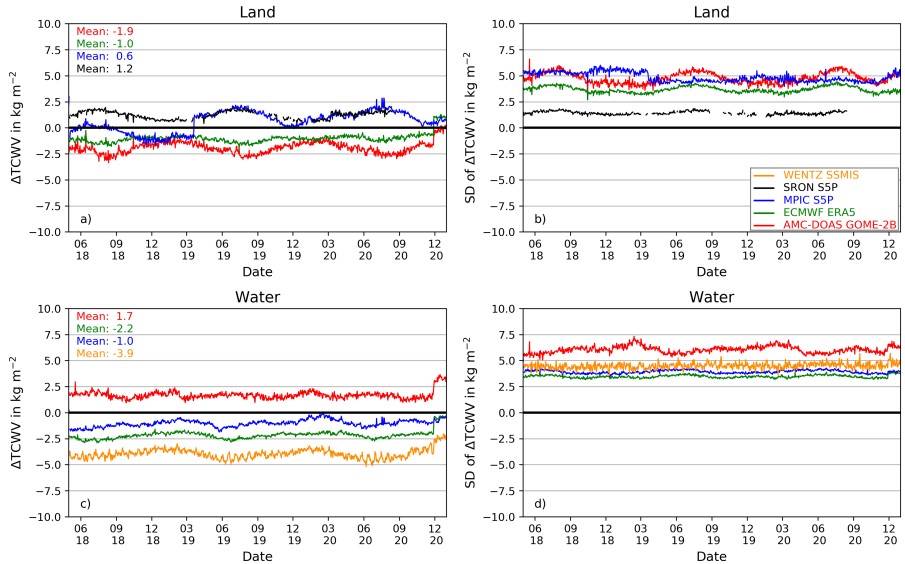

**Figure 7.** Time series of the daily averaged difference between TCWV$_{\text{AMC,S5P}}$ and TCWV from other data sets over water (a) and land (c) surface from May 2018 to December 2020. The right panel shows the respective standard deviation over land (b) and water (d).

referred to as the clear sky bias (Gaffen and Elliott, 1993; Sohn and Bennartz, 2008). In TCWV$_{\text{AMC,S5P}}$ data with cloud fractions above 0.2 are excluded, which causes a negative offset.

– The AMC-DOAS retrieval and also the cloud and albedo correction for S5P use as reference a tropical profile. Usually the reference profile shape and the true profile shape differ. That also can cause systematic deviations especially in the presence of remnant clouds.

– The cloud and albedo treatment is dependent on the quality of the used cloud products. Uncertainties in the cloud product will have an impact on the surface albedo estimation and also on the calculation of the correction factors.

There also are some values where TCWV$_{\text{WENTZ,SSMIS}}$ exceeds TCWV$_{\text{AMC,S5P}}$ by more than $20 \, \text{kg} \, \text{m}^{-2}$. These arise from two DSMP F16 orbits located between the International Date Line and North and South America. Those orbits belong to the very first orbits of the daily SSMIS TCWV product. For the TCWV$_{\text{AMC,S5P}}$ the transition between the first and last orbit of the specific day is located much closer to the International Date Line. This results in a time mismatch of roughly 24 h between S5P data and SSMIS data in these areas causing these observed TCWV differences.

## 4.2 Time series of TCWV differences

Since TCWV$_{\text{AMC,S5P}}$ is available for more than two years it is worthwhile to investigate the behaviour of differences of TCWV throughout the time. Fig. 7 shows the daily averaged TCWV differences between the TCWV$_{\text{AMC,S5P}}$ product and the different





correlative data sets from May 2018 to December 2020. Again this is done for land surface (Fig. 7a) and water surface (Fig. 7c) separately. The respective standard deviations are also shown (Fig. 7b,d)

The temporal behaviour of the TCWV difference between the AMC-DOAS products for S5P and GOME-2B over land shows a systematic deviation between -1 kg m$^{-2}$ -2.5 kg m$^{-2}$; also, a seasonal cycle is visible. The largest deviations can be

seen during the northern summer months whereas in northern winter the deviation is least. The standard deviation also shows a seasonal cycle with largest variability also during northern summer. Compared to the other data sets the average difference is largest.

The behaviour of the difference to the S5P AMC-DOAS product is quite different for S5P data set from MPIC. There is a clear jump of around 1.8 kg m$^{-2}$ located at 20 March 2019. This is due to an update of the TROPOMI FRESCO-S cloud

product used in the generation of TCWV$_{MPIC,S5P}$, which affects the retrieved TCWV. Before the jump the deviation is between 0.5 kg m$^{-2}$ and -1 kg m$^{-2}$, hereafter the deviation is slightly positive. This jump mainly originates from the tropical regions where evergreen rainforests are common, e.g. Amazon rainforest.

The difference also shows a seasonal cycle which is of opposite sign compared to the difference between the two AMC-DOAS data sets. The seasonal cycle cannot be observed in the standard deviation. The standard deviation of $\Delta$TCWV$_{AMC,S5P-MPIC,S5P}$

reveals a reduction of about 1 kg m$^{-2}$ caused by the effects of the change in the FRESCO-S cloud product.

The difference between TCWV$_{AMC,S5P}$ and TCWV$_{SRON,S5P}$ shows a deviation of around 1.2 kg m$^{-2}$ and also a seasonal cycle with least differences in northern winter and largest differences during northern summer. This is similar to the seasonality of $\Delta$TCWV$_{AMC,S5P-MPIC,S5P}$. After March 2019 the deviation to both S5P TCWV products are also very similar. At 7 March 2020 there was a change in the cloud product used by SRON. As for $\Delta$TCWV$_{AMC,S5P-MPIC,S5P}$ this causes a small jump in

$\Delta$TCWV$_{AMC,S5P-SRON,S5P}$ of around 0.5 kg m$^{-2}$. The standard deviation varies around 1.5 kg m$^{-2}$, which is lower compared to other TCWV products. The TCWV$_{SRON,S5P}$ are filtered with a very strict cloud filter that only left small TCWV values. Therefore TCWV values from tropical regions where TCWV is high are discarded.

Between TCWV$_{AMC,S5P}$ and TCWV$_{MOD,ERA5}$ there is a general negative deviation around 1.6 kg m$^{-2}$. In contrast to the other data sets only very small seasonal variability can be seen.

In summary, the mean deviation between TCWV$_{AMC,S5P}$ and other TCWV data ranges from -1 to -4 kg m$^{-2}$. The smallest difference can be observed between TCWV$_{AMC,S5P}$ and TCWV$_{MPIC,S5P}$. The deviation TCWV$_{WENTZ,SSMIS}$ to TCWV$_{AMC,S5P}$ ranging from -3 to -4 kg m$^{-2}$ is largest also throughout the time. With except of the difference between the two AMC-DOAS products there is also a small seasonal feature in $\Delta$TCWV with largest deviations between TCWV$_{AMC,S5P}$ and other data products in January and smallest deviation in July. These deviations for different seasons are also of similar magnitude. In

contrast to land and also to other data sets over ocean the mean deviation between TCWV$_{AMC,S5P}$ and TCWV$_{AMC,GOME-2B}$ is positive without seasonal features. The standard deviation of the TCWV differences ranges between 3 to 5 kg m$^{-2}$ with except for $\Delta$TCWV$_{AMC,S5P-AMC,GOME-2B}$ which is around 6 to 7 kg m$^{-2}$.

At the end of November 2020 there was a version change in the FRESCO product which is used to correct for cloud effects and is also used to calculate surface albedo to derive the AMC-DOAS S5P TCWV product. This caused a general increase



**Table 3.** Global average ($\pm$ standard deviation) of the TCWV products and average difference to the AMC-DOAS S5P TCWV product ($\Delta$TCWV) for January 2019 in kg m$^{-2}$.

| | Land | | Sea | | Global | |
|---|---|---|---|---|---|---|
| Dataset | Mean | $\Delta$TCWV | Mean | $\Delta$TCWV | Mean | $\Delta$TCWV |
| AMC-DOAS S5P | 11.6$\pm$13.8 | – | 22.5$\pm$15.3 | – | 18.7$\pm$15.7 | – |
| ERA5 | 12.1$\pm$15.3 | -1.0$\pm$2.5 | 24.3$\pm$15.2 | -1.8$\pm$1.5 | 20.0$\pm$16.3 | -1.5$\pm$2.0 |
| AMC-DOAS GOME-2 | 12.0$\pm$14.3 | -1.1$\pm$2.8 | 20.7$\pm$13.9 | 1.7$\pm$2.8 | 17.5$\pm$14.6 | 0.7$\pm$3.1 |
| MPIC S5P | 19.8$\pm$16.6 | -1.7$\pm$4.3 | 24.5$\pm$15.7 | -1.8$\pm$2.2 | 23.3$\pm$16.0 | -1.8$\pm$2.8 |
| SRON S5P | 8.2$\pm$9.3 | 1.4$\pm$2.5 | – | – | 8.2$\pm$9.3 | 1.4$\pm$2.5 |
| SSMIS | – | – | 27.2$\pm$15.4 | -4.8$\pm$2.3 | 27.2$\pm$15.4 | -4.8$\pm$2.3 |

**Table 4.** Same as Tab. 3, but for July 2019.

| | Land | | Sea | | Global | |
|---|---|---|---|---|---|---|
| Dataset | Mean | $\Delta$TCWV | Mean | $\Delta$TCWV | Mean | $\Delta$TCWV |
| AMC-DOAS S5P | 22.7$\pm$11.7 | – | 23.3$\pm$15.5 | – | 23.1$\pm$14.5 | – |
| ERA5 | 24.5$\pm$13.2 | -1.9$\pm$2.9 | 24.9$\pm$15.6 | -2.2$\pm$1.8 | 24.8$\pm$14.9 | -2.1$\pm$2.2 |
| AMC-DOAS GOME-2 | 24.9$\pm$12.6 | -2.0$\pm$3.7 | 21.5$\pm$14.3 | 1.7$\pm$2.8 | 22.5$\pm$13.9 | 0.6$\pm$3.5 |
| MPIC S5P | 21.2$\pm$11.8 | 1.7$\pm$3.3 | 26.0$\pm$16.0 | -2.2$\pm$2.5 | 24.5$\pm$15.0 | -1.0$\pm$3.3 |
| SRON S5P | 18.5$\pm$8.6 | 2.0$\pm$3.3 | – | – | 18.5$\pm$8.6 | 2.0$\pm$3.3 |
| SSMIS | – | – | 28.8$\pm$16.4 | -5.3$\pm$2.3 | 28.8$\pm$16.4 | -5.3$\pm$2.3 |

of 2.3 kg m$^{-2}$ in TCWV$_{AMC,S5P}$ over both land and water surfaces. Due to this increase the deviations also show this jump of around 2 kg m$^{-2}$ except for $\Delta$TCWV$_{AMC,S5P-MPIC,S5P}$.

## 4.3 Assessment of the spatial dependence of the difference

To investigate possible reasons for e.g. the seasonal cycle or the different temporal behaviour among the data sets we present monthly mean global maps of all data and their difference to our new product. The monthly comparison is restricted to January and July 2019 because all typical spatial and temporal features can already be seen from these months. Values for the global average of the TCWV products and its standard deviation and also their difference to TCWV$_{AMC,S5P}$ can be found in Tab. 3. for January and Tab. 4 for July.



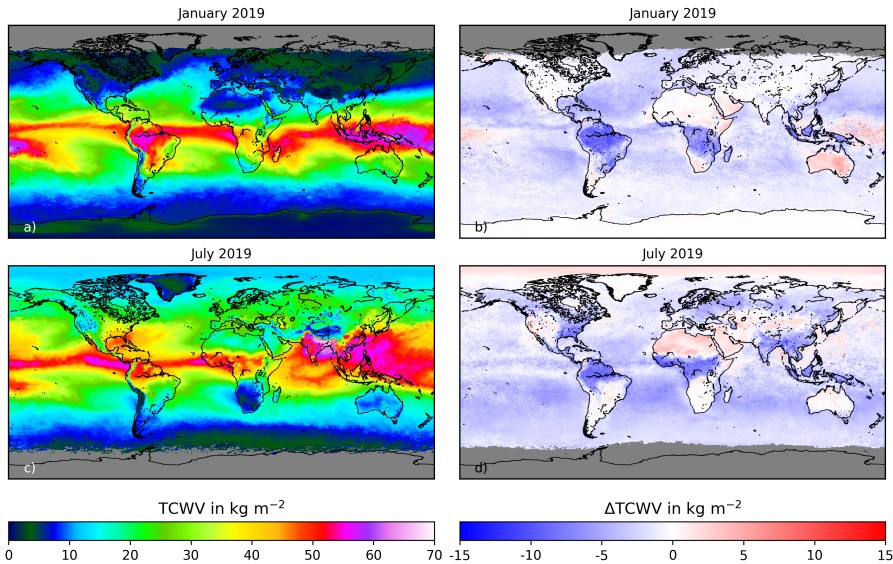

**Figure 8.** Global map of monthly averaged ERA5 TCWV (left) and the difference AMC-DOAS S5P TCWV – ERA5 TCWV (right) for January and July 2019 (top, bottom).

### 4.3.1 ERA5

Fig. 8 shows a comparison of the AMC-DOAS S5P TCWV products with ERA5 model data. The temporal and spatial sampling of the ERA5 data is the same as for TCWV$_{\mathrm{AMC,S5P}}$, because we selected the closest model data point for each S5P measurement. All spatial features of the differences can be seen in Fig. 8b,d.

Overall, the ERA5 TCWV is slightly larger. On global average the deviation varies around -1.5 kg m$^{-2}$ with a standard deviation of 2.0 kg m$^{-2}$ for January. In July there is a larger deviation of -2.1 kg m$^{-2}$ and also a larger standard deviation of 2.2 kg m$^{-2}$ which originates from features over land. Over land the deviation is around -1.0 kg m$^{-2}$ with a variability not larger than 2.5 kg m$^{-2}$. The tropical region contributes most to the negative deviation in all time periods. In July, the negative deviation spreads northward. The large tropical deviations are collocated with the evergreen rain forests in the Amazon region,

central Africa and also the tropical islands of Asia. July is vegetation growth season in the northern hemisphere, i.e. the trees built up leaves. This pattern reveals potential influences of vegetation either on the TCWV$_{\mathrm{AMC,S5P}}$ or on the FRESCO product which is used as input for the correction. Other features can also be seen in large parts of central Australia in January and Sahara in July. Both regions are deserts; they show positive deviations up to 5 kg m$^{-2}$ during local summer months. Despite of both features the discrepancies between TCWV$_{\mathrm{AMC,S5P}}$ and TCWV$_{\mathrm{MOD,ERA5}}$ over land are very close to zero over large regions.

Over sea the deviations ranging from -1.8 kg m$^{-2}$ to -2.2 kg m$^{-2}$ are larger than over land by 0.3 kg m$^{-2}$ in July and around 0.8 kg m$^{-2}$ during January. The distribution of the deviation is quite homogeneous denoted by the reduced standard deviation of around 1.5 kg m$^{-2}$. Within the 30 °S and equator band the deviation is slightly larger than in other regions in July.





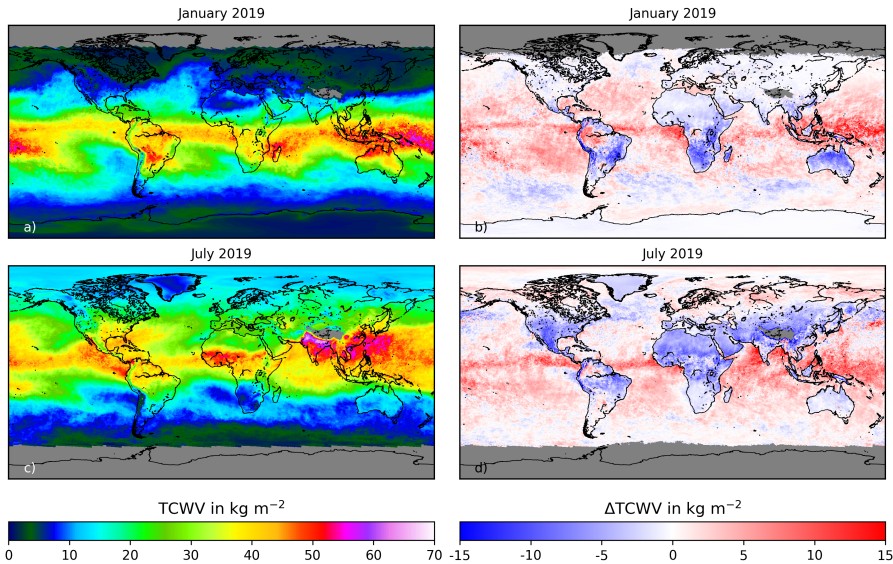

**Figure 9.** As Fig. 8, but for the AMC-DOAS GOME-2B TCWV product.

### 4.3.2 GOME-2B

In general, the monthly averaged $TCWV_{AMC,GOME-2B}$ (Fig. 9a,c) shows the same features as $TCWV_{AMC,S5P}$ (Fig. 5). The main distinctions to $TCWV_{AMC,S5P}$ are evident over the tropics where $TCWV_{AMC,GOME-2B}$ barely exceeds $50\,kg\,m^{-2}$. There are no $TCWV_{AMC,GOME-2B}$ data over Himalayan mountain ranges due to exclusion of GOME-2 values when the air mass correction is too large.

More details are revealed by the difference maps (Fig. 9b,d). As can be seen there are systematic spatial structures in the differences. In general, the highest differences both over land and water occur close to the tropics where absolute TCWV is high. In the mid-latitudes and polar regions the deviations are close to zero. Over land surface negative deviations are prevailing, e.g. over entire Africa or India. Especially over Africa several effects can be seen. In the northern part where the surface is bright due to the deserts the albedo correction reduces the $TCWV_{AMC,S5P}$ resulting in a difference of around $-5\,kg\,m^{-2}$ especially in July. The southeastern part of Africa also shows enhanced differences. This region is typically more elevated than the northeastern part. This difference in the surface elevation is only considered in the $TCWV_{AMC,S5P}$ product, which results in lower TCWV over mountain regions and thus explains the larger retrieved $TCWV_{AMC,GOME-2B}$ (which represents the column from sea level) there. These differences are therefore mainly due to the definition differences between both data sets. In July the deviations over land are largest and widely spread throughout the northern hemisphere due to overall increase of TCWV.

Over sea the difference is overall positive. Since over ocean the surface height is zero, the differences between the two data products cannot be attributed to the different TCWV definitions, they have to be related to the postprocessing, which is





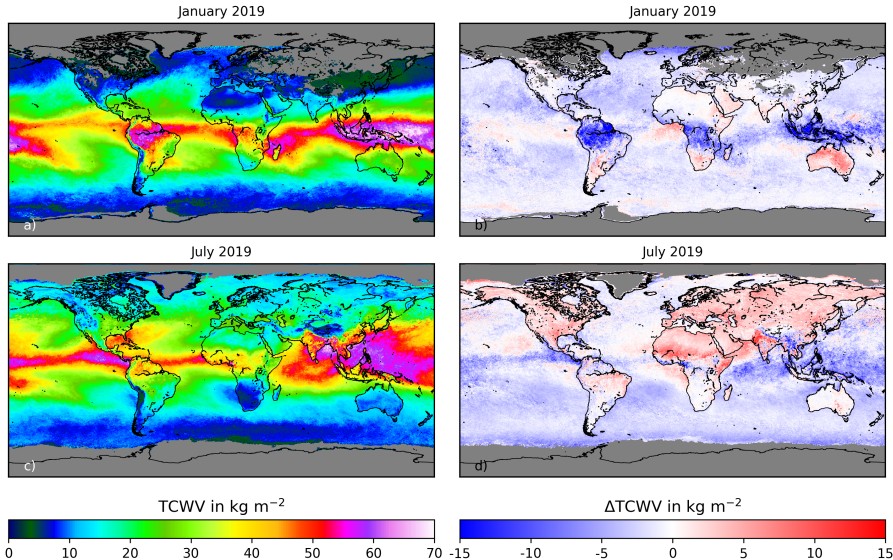

**Figure 10.** As Fig. 8, but for the S5P TCWV product from MPIC.

only performed for the S5P product. The mean sea surface albedo does not deviate much from the assumptions made in the retrievals, therefore differences caused by the post-processing corrections are more likely to be related to clouds.

Most deviations can be seen in the tropical area where the average TCWV is largest. The ITCZ, where highest TCWV occurs, is also predominated by clouds due to enhanced convection. Here, the correction of cloud effects affects the already
high TCWV in the tropics more than in other regions.

The difference of the crossing time of GOME-2 on MetOp-B and Sentinel-5p at equator is about four hours. This also can have an effect due to diurnal cycles in TCWV and in cloud cover. In some areas like the stratocumulus cloud shields over ocean there is a diurnal cycle with enhanced cloud cover in the morning hours and decreased cloud cover during the afternoon hours (Noel et al., 2018). This may reduce the retrieved $TCWV_{AMC,GOME-2B}$ due to more cloudiness during the morning overpass of
GOME-2 on MetOp-B. Over land the situation is reversed due to more pronounced convective clouds in the afternoon hours.

The daily comparison between $TCWV_{AMC,S5P}$ and $TCWV_{AMC,GOME-2B}$ discussed earlier showed a quite large standard deviation of about 6 to $7\,kg\,m^{-2}$. This can be related to the spatial structure of the deviation over the sea surface whereby largest discrepancies are located in the tropics.

### 4.3.3   MPIC S5P

The S5P TCWV data sets provided by MPIC gives the opportunity to compare different methods applied to the same instrument. This reduces the effect of possible temporal changes of TCWV as a source of uncertainty in the comparison results. The averaged $TCWV_{MPIC,S5P}$ (Fig. 10a,c) shows similar structures as $TCWV_{AMC,S5P}$ (see Fig. 5 for comparison). Over the western Pacific close to Indonesia there are values up to $70\,kg\,m^{-2}$ which are not shown in the averaged $TCWV_{AMC,S5P}$.





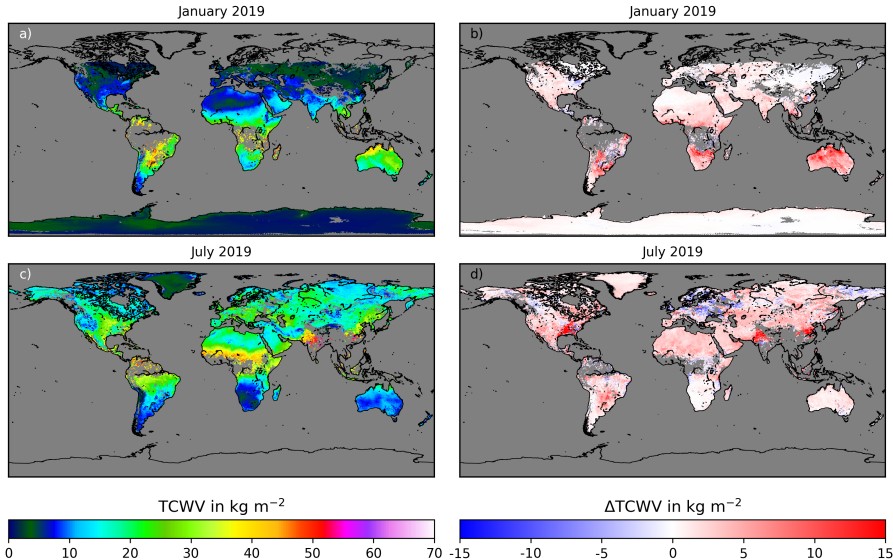

**Figure 11.** As Fig. 8, but for the S5P TCWV product from SRON.

Main discrepancies between $TCWV_{AMC,S5P}$ and $TCWV_{MPIC,S5P}$ (Fig. 10b,d) occur over land in tropical regions (Amazonas, Indonesia, central Africa) during January. As mentioned before, there was a jump in the daily averaged difference between both data sets at 20 March 2019 due to changes in cloud parameters used by MPIC. Before this change $\Delta TCWV_{AMC,S5P-MPIC,S5P}$ is enhanced in these regions (see Fig. 10b). In July (Fig. 10d), $\Delta TCWV_{AMC,S5P-MPIC,S5P}$ does not show such large values as in

January anymore.

In other areas there are far less deviations. Australia shows larger positive deviations during January but also in South America and in southern parts of Africa positive deviations are observed. During July all land masses on the northern hemisphere show constant positive deviations whereas south of the equator there are negligible differences.

Over sea there is an overall negative difference which is slightly larger in July (-2.2 kg m$^{-2}$) than in January with -1.8 kg m$^{-2}$.

Large areas of negative deviations can be observed in the southern hemisphere during July whereas in the northern hemisphere deviations are very close to zero. A reversed pattern is slightly visible in January. Largest discrepancies can be seen close to Indonesia. The ITCZ is also marked by the narrow band of slightly negative deviation. This can be due to different cloud correction schemes.

### 4.3.4   SRON S5P

Fig. 11a,c shows the $TCWV_{SRON,S5P}$. The data gaps within the tropics are clearly visible. Despite monthly averages there are no data within one month there. This is related to the strict cloud filter, because especially the tropics are associated with higher cloud occurrence.





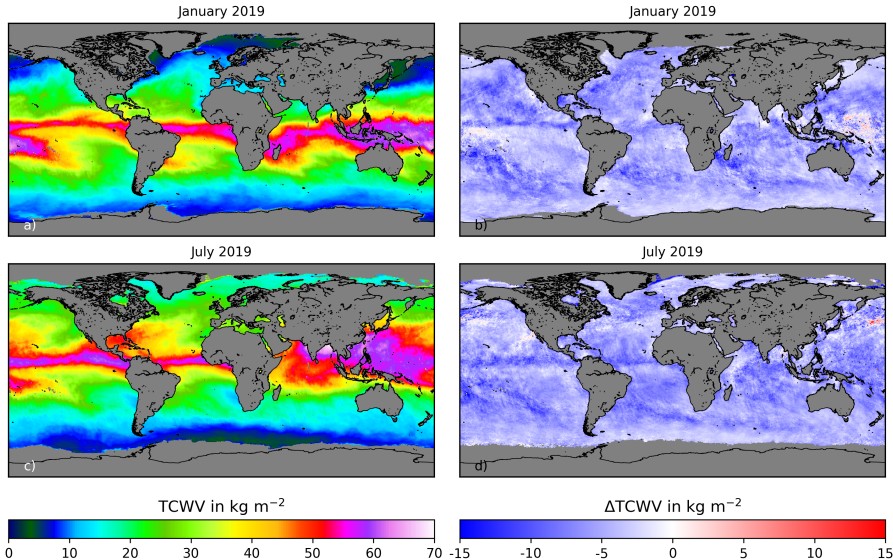

**Figure 12.** As Fig. 8, but for the SSMIS TCWV product.

The difference to the monthly averaged AMC-DOAS S5P data ($\Delta\text{TCWV}_{\text{AMC,S5P-SRON,S5P}}$) is mainly positive during January (Fig. 11b). On average, the deviation is $1.4\,\text{kg}\,\text{m}^{-2}$ for this month. Especially southern hemispheres land masses except for the Antarctic show quite large positive deviations up to more than $10\,\text{kg}\,\text{m}^{-2}$ during January. The largest differences can be observed in the northern part of Australia. The northern hemisphere shows only small and spatial homogeneous positive deviations during January.

In July the land masses on the southern hemisphere shows positive deviation over South America whereas Australia and South Africa the difference between both S5P TCWV products is very close to zero. Some spots with large positive deviation also can be seen in parts of the United States of America, northern India and also parts of China. These areas suffer from poor data availability of $\text{TCWV}_{\text{SRON,S5P}}$ such that the average only consists of few days. The averaged difference is slightly larger in July with $2.0\,\text{kg}\,\text{m}^{-2}$.

Note that in many cases the $\text{TCWV}_{\text{SRON,S5P}}$ monthly averages consists of not more than five days or less. Actually, only over deserts there is more than the half of daily data available within a month. This sampling difference between the products may explain some of the observed deviations.

### 4.3.5 SSMIS

The spatial distribution of $\text{TCWV}_{\text{WENTZ,SSMIS}}$ is shown in Fig. 12a,c. There are no data over land surfaces and also not over sea ice. The averages over sea are around $27\,\text{kg}\,\text{m}^{-2}$. The deviation (Fig. 12b,d) to $\text{TCWV}_{\text{AMC,S5P}}$ show an overall negative deviation of around $-5\,\text{kg}\,\text{m}^{-2}$ which is more than for the other data sets. There are also structures visible, e.g. a tongue of slightly more enhanced discrepancy located over southern parts of the Pacific.





This deviation can be due to sampling differences as discussed earlier. The time difference of more than 2h also can have an effect on the $\Delta\text{TCWV}_{\text{AMC,S5P-WENTZ,SSMIS}}$.

## 5 Conclusions

The AMC-DOAS approach was successfully applied to S5P measurements to detect TCWV. For this purpose, several im-
provements of the retrieval method have been developed. This includes an update of the underlying radiative transfer data base which now especially also considers variable surface elevation. Due to the latter, the AMC-DOAS product is now defined as the TCWV relative to the surface whereas it was defined relative to sea surface before. This especially results in on average lower TCWV values over land.

In addition to the usually applied filtering based on the derived air mass correction factor and solar zenith angle also new
filters are applied. These use the S5P FRESCO cloud fraction and cloud height relative to the surface.

Additional post-processing procedures have been established to account for variable surface albedo and remnant clouds. Furthermore, an empirical correction has been developed and applied which reduces systematic striping structures in the retrieved TCWV over ocean. The origin of these structures is currently unclear. They are assumed to be instrumental features, but this issue needs further investigations.

Except for the empirical stripes correction, the newly developed algorithm modifications are instrument independent and may thus also be applied to GOME, SCIAMACHY and GOME-2 to further improve also these AMC-DOAS TCWV data products.

The updated AMC-DOAS retrieval has been applied to all S5P measurements from May 2018 to December 2020 which results in a new global TCVW data set. This product was validated by comparison with various independent data sets, namely
with the GOME-2B AMC-DOAS product, ECMWF ERA5 model data and the MPIC S5P TCWV product over land and ocean and with SSMIS data over ocean.

The new S5P AMC-DOAS TCWV data agree reasonably well with these other data sets within about $\pm 2.5\,\text{kg}\,\text{m}^{-2}$ except for the SSMIS product which shows about two times larger negative deviations. Differences to ERA5 model data and the S5P TCWV product from MPIC show more negative values over sea than land. That indicates a small land sea bias of typically
not more than $1\,\text{kg}\,\text{m}^{-2}$ of the AMC-DOAS S5P TCWV. Best agreement was achieved by comparing S5P TCWV from MPIC and AMC-DOAS S5P TCWV. Largest discrepancies between the AMC-DOAS products and the product from ERA5 are found over regions with large vegetation within the growth season. Between the two AMC-DOAS data sets for S5P and GOME-2B small offsets were visible with positive deviation over sea and negative deviation over land which are mainly caused by the changes in the AMC-DOAS retrieval applied to S5P. Small seasonal cycles can be found over land and sea.

The standard deviation of the differences is similar for all data sets and lies in the range 3-5 $\text{kg}\,\text{m}^{-2}$ except for the GOME-2B AMC-DOAS product over sea which varies by up to 7 $\text{kg}\,\text{m}^{-2}$ due to the postprocessing applied to AMC-DOAS S5P TCWV. Another exception is the variability of the difference between AMC-DOAS S5P TCWV and the S5P TCWV from SRON of around 1.5 $\text{kg}\,\text{m}^{-2}$ which is lower than for the other data sets due to the much stricter filtering of SRON S5P TCWV.



The observed standard deviations are in agreement with comparison studies for the existing AMC-DOAS products (see e.g. Noël et al., 2005; Kalakoski et al., 2016) which show a typical scatter of around $5\,\mathrm{kg\,m^{-2}}$. This variability arises from different overpass times resulting in systematic changes in the atmospheric conditions due to e.g. transport processes and altering cloud cover. The value of $5\,\mathrm{kg\,m^{-2}}$ can therefore be considered as a rough estimate for the natural variability of TCWV in

combination with effects from different spatial and temporal sampling. All TCWV data sets used in this study show systematic differences between each other which are typically smaller than this.

Especially the comparisons of the different S5P results show that there is no "best" algorithm or product for TCWV; all retrieval methods / spectral regions have their advantages and disadvantages. The dynamic range of the TCWV in the atmosphere benefits from having a variety of approaches to measure this quantity, thus complementing each other. The large variability of

atmospheric water vapour requires a variety of approaches, which can complement each other.

The parameters used here to compare AMC-DOAS TCWV and other TCWV products shows similar values which are also found by (e.g. Van Malderen et al., 2014; Schröder et al., 2016; Schröder et al., 2018).

These studies also reveal that typical differences around $5\,\mathrm{kg\,m^{-2}}$ occur when comparing different TCWV data sets. The current AMC-DOAS S5P TCWV product relies on FRESCO input data for the albedo / cloud correction and for filtering.

Therefore, changes in the input cloud product can have an effect on the derived TCWV. This problem can be seen e.g. in the jump observed in the S5P product from MPIC, which originates in an algorithm change of the used cloud product. AMC-DOAS S5P TCWV also shows a general increase of on average more than $2\,\mathrm{kg\,m^{-2}}$ at the end of November. This is caused by a version change of the FRESCO cloud product. It is planned to investigate possibilities to retrieve the required cloud properties independently from external data, e.g. by a FRESCO-like cloud detection scheme from the oxygen B band. This would make

the AMC-DOAS retrieval method even more independent from external data sets.

In summary, the AMC-DOAS method has proved to be a powerful and fast tool to retrieve TCWV from large data sets. The application to TROPOMI/S5P data provides spatially highly resolved results, which allows to investigate very small scale features in the TCWV.

*Data availability.* The AMC-DOAS TCWV products for S5P and GOME-2 are available on request from the authors. The MPIC TCWV

are also available on request from the authors. The SRON S5P H2O version 9_1 data are available under ftp://ftp.sron.nl/open-access-data-2/TROPOMI/tropomi/hdo/9_1/. The SSMIS data are available at http://www.remss.com/missions/ssmi/.

The ERA5 reanalysis data were obtained directly from ECMWF but are also available from the Copernicus Atmosphere Monitoring Service (https://www.doi.org/10.24381/cds.bd0915c6, last access 21 April 2021)

*Author contributions.* Tobias Küchler applied the AMC-DOAS retrieval (including postprocessing) to S5P data. Stefan Noël developed the

AMC-DOAS method and provided also the AMC-DOAS GOME-2B TCWV data. Andreas Schneider and Tobias Borsdorff produced and provided the SRON S5P TCWV data. Thomas Wagner and Christian Borger provided MPIC TCWV data for S5P. All authors including Heinrich Bovensmann and John P. Burrows contributed to the preparation of the manuscript.



*Competing interests.* Thomas Wagner is the executive editor of AMT.

*Acknowledgements.* SSMIS data are produced by Remote Sensing Systems. Data are available at www.remss.com/missions/ssmi. We thank ESA for provision of Level 1 data and Level 2 FRESCO data from Sentinel-5p. The ERA5 data are provided by European Centre for Medium Range Forecasts. The SRON TROPOMI data processing was carried out on the Dutch national e-infrastructure with the support of the SURF Cooperative. GMTED2010 data are provided by the U.S Geological Survey.

Special thanks goes to the University of Bremen which funded this work. All calculations reported here were performed on HPC facilities of the IUP, University of Bremen, funded under DFG/FUGG grant INST 144/379-1 and INST 144/493-1.





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
