# Peer review of "Total water vapour columns derived from Sentinel 5p using the AMC-DOAS method"

_Atmospheric Measurement Techniques, 2021_

## Author Comment (AC1)

The paper by Hummel et al on the

"Total water vapour columns derived from Sentinel 5p using the AMC-DOAS method"

provides a detailed discussion of the results of total column water vapour (TCWV) retrievals from the Sentinel-5p TROPOMI instrument using the AMC-DOAS retrieval methods originally developed by Noel et al.

The results are compared in particular to the most recent ECMWF ERA5 reanalysis model data dominated by thermal infrared, GPS, and radiosonde water vapour profile data assimilation, as well as to other, model independent, retrieval results, in particular MPIC DOAS operating in the UV region, AMC-DOAS TCWV results from the GOME-2 instrument on Metop, and TCWV results from the SSMIS microwave sensor.

Additional corrections on the impact on surface albedo and elevation, as well as empirical corrections, potentially due to instrument artefacts, have been applied to the AMC-DOAS retrieval results from S5p.

The authors find good agreements between all data-sets with typical standard-deviations expected for any TCWV retrieval validation, due to differences in spatial scales and resolutions, as well as low mean biases, in particular also to the ERA5 model.

This is in contrast to previous studies where comparison to reanalysis model data revealed larger biases, typically also due to statistical effects of comparing "cloudy" model results with typically cloud free or low cloud) retrieval results. This fact points to significant improvements in the reanalysis data modelling of small-scale relative humidity and clouds, however not discussed by the authors. Also in this respect to the paper may serve as a good validation of the ERA5 re-analysis data, in particular in regions where assimilated data in the reanalysis fields is sparse, like in remote areas over land, and TCWV from thermal infrared missions biased because of lack of thermal contrast.

The impact of the additional corrections applied to S5P version of AMC-DOAS is significant, in particular with respect to surface elevation, albedo and clouds. While the elevation related effects on the TCWV are obvious and discussed when comparing to the "non-corrected" Metop-B AMC-DOAS results, the discussion on the additional albedo and cloud effects are less conclusive. But the authors expect that the same correction should be principally always applied, because they are basically instrument independent, and therefore would also apply for the GOME-2 data-set. This should indeed be followed up in future work.

The paper is well written and I can recommend it for publication in AMT.

The general introduction is however missing an overall conclusion with respect to the significance of the TCWV products derived from the UV-visible to NIR spectrum, in addition to the large amount of data-sets on TCWV and profile information existing and extensively used from the thermal infra-red and MW region. This is important in the context of a longer discussion on the application range of these data-sets, linking back to the discussion referred to above. The relevance of these visible data-sets is their coverage and (relatively) weaker dependence on surface type, as well as their sensitivity to the surface WV profile layers, ie. their accuracy w.r.t. the total column.

Being independent of model re-analysis data, they can be used for the validation of the latter. It would improve the motivation of why an operational WV product from S5p, as well as from S5 should be made available in the future.

Thank you for your recommendation for publication on AMT. We will include a short summary about the significance of the TCWV products:

„In general, water vapour products derived from the visible (VIS) to the near infrared (NIR) spectral range have the advantage that the measurements are sensitive to the surface. They have a relatively weaker dependence on surface type than products from other wavelength regions and usually cover both land and ocean. Specifically, the AMC-DOAS retrieval method does not rely on external datasets and therefore provides completely independent water vapour data. The independence of the VIS-NIR water vapour products of model re-analysis data makes them useful for the validation of the latter. "

Specific comments:

Section 3.4 on the empirical correction:

(Q1) There seems no indication (from experience from other retrievals?) of what causes the artefacts. Usually such across track features (which are frequently observed for such data, also in other VIS/NIR spectrometer missions) can be typically the result of an additive offset: i.e. stray-light, or a problem in the solar irradiance spectrum. What exactly is used for the latter? Experimenting with I0 (using artificial offsets or a very low TWVC (polar region) reference spectrum) potentially can reveal the origin.

(A1) For almost every day there is a irradiance measurement from S5P. When calculating TCWV from the S5P orbits we use the latest available irradiance measurement. If there is no irradiance measurement for one specific day we use that one from the day before.

Straylight in either the solar or the earthshine data might indeed be a potential reason for the artefacts which is worth further investigations. We will consider the reviewer's suggestions for a future product version.

(Q2) Can it be that the surface albedo correction already takes out a significant part of the feature over land? Since, if the artefact is not visible over land it can hardly be an instrumental feature.

(A2) The feature is not visible over land even before the albedo correction is done. The effect is mainly visible over ocean . This does not exclude an instrumental feature: For example, a straylight offset could have a larger effect over ocean where the surface albedo and therefore the signal is low. Other possible reasons for this effect could be aerosols, or angular dependence of the surface reflection which also might  have different/larger impact over ocean. We will discuss this in the updated paper.

(Q3) Section 4.3, on the assessment of spatial features:

It should be generally not be surprising to find a feature related to vegetation in the AMC DOAS retrievals given is usage of measurements from the red-edge of the spectrum. The behaviour of MPIC DOAS in this respect is not surprisingly much different. Reducing albedo variations was in the end a large motivation I would assume (not only higher albedo over water as stated in the paper, and spectral regions covered by future instrumentation)

to move towards the blue. This specific sensitivity of a TCWV retrieval operation around 700 nm should be better highlighted and discussed in the paper.

(A3) We agree that the red edge causes some  sensitivity of the retrieval to vegetation; however, since we use a very small fitting window most of this is handled via the polynomial fit. What we see is actually a second order effect. Note that the MPIC product also has some sensitivity on vegetation although it is based on the measurements in the blue spectral range. We will explain this in the updated paper. The following will be added to the conclusion section:

„This pattern reveals potential influences of vegetation either on the AMC-DOAS S5P TCWV or on the FRESCO product which is used as input for the correction.This could be related to the spectral variation of the surface albedo („red edge") which is not fully captured by the polynomial fit."

Editorials:

Introduction, l.21: Full stop missing.

Section 3.3.1, l9f: Sentence should be improved.

We will replace „This is the reason why we limit from present the scenes being studied to those having cloud fractions from 0 to 0.2"

to

„To reduce the impact of clouds we therefore filter out scenes with cloud fractions larger than 0.2 as a first step."

Acknowledgments: Acknowledgment to EUMETSAT providing GOME-2B level-1 data missing.

We will address the issues mentioned above in the revised version.

Thank you very much for your constructive feedbacks and questions.

---

## Author Comment (AC2)

Total water vapour columns derived from Sentinel 5p using the AMC-DOAS method, by Tobias Kuechler et al.

The paper describes the application of the existing AMC-DOAS algorithm, for the retrieval of total column water vapour, to measurements of the TROPOMI instrument on Sentinel-5P. Furthermore, an inter-comparison is carried out with other data sets.

These other data are partly also based on TROPOMI data, and partly independent.

As the AMC-DOAS algorithm has previously been applied to other satellite instruments, there are no big surprises here.

However, some algorithm improvements were made, notably albedo correction and cloud filtering, that clearly improve the results.

The paper serves as a useful description of the improved algorithm and the AMC-DOAS data product from S5P.

In general, the paper is well written but there are several issues that need clarification and/or more discussion.

See my detailed comments below.

The paper is appropriate for publication in AMT once these issues have been addressed.

Thank you for recommending the paper for publication.

Detailed comments

(Q1) in general: the use of the word "ground pixel" is confusing. A ground pixel may be anywhere on the Earth surface. It seems you mean across-track [ground] pixel or viewing angle. Please replace or clarify in the text.

(A1) The term „ground-pixel" refers to the across track pixel of the TROPOMI instrument. We will change this in the article.

(Q2) p.8 Eqation(1) indicates that no additional cross-sections are used for Ring, spectral surface albedo (vegetation, chlorophyl) and liquid water. Please give a rationale.

(A2) The polynomial P in our AMC-DOAS equation handles a lot of broadband effects. Since our fitting window is small, this should on first order also cover effects of spectrally varying surface albedo or liquid water absorption. However, there may be second order effects (like the larger offsets to ERA-5 we see over vegetation). Wagner et al. (2009) („Three-dimensional simulation of the Ring effect in observations of scattered sun light using Monte Carlo radiative transfer models") stated: towards longer wavelenths the propability of inelastic scattering decreases to values lower than 1%. We will add this as follows:
„As in standard DOAS, P is a low order polynomial accounting for broadband features such as scattering, liquid water absorption and also variations in the surface albedo. The Ring effect is not considered because it is not relevant in our fit window."

(Q3) p.9 line 4. Is the viewing zenith angle (across-track pixel number) not used in the radiative transfer ?

(A3) Within the RTM simulations the viewing zenith angles are not used. This is not necessary, because the AMF correction handles different viewing geometries. We will mention this in the paper as follows:

„Effects due to different observation geometries are also accounted for by *a*. Varying viewing angles are therefore not considered within the radiative transfer model."

(Q4) p.9 line 19. The "default" albedo of 0.02 is very low for land surfaces. Mention (later in the text) that this often may result in a large albedo correction.

(A4) Indeed, the „default" albedo of 0.02 is representative for ocean and too low for land.. However, after correcting albedo and cloud effects the AMC-DOAS TCWV does no longer depend on the choice of the „default" surface albedo, although of course corrections are larger over land. We will mention this in the upated text as follows:

„Larger surface albedo is associated with an overestimation of the reference TCWV and a correction factor smaller than one. This is especially true over bright land surfaces such as snow covered areas and deserts where the the AMC-DOAS TCWV correction would be large. However, after correcting albedo and cloud effects the AMC-DOAS TCWV does no longer depend on the choice of the „default" surface albedo."

(Q5) Please discuss if the limit of 0.8 on the airmass correction factor (p.9 line 28) or the limit of >0.6  <1.2 on the albedo/cloud correction factor (p.13 line 13) can cause a land/sea bias in partially clouded scenes, because over land you "always" have an offset to the correction factor compared to sea.

(A5) Indeed, the AMCF can be higher over land due to different conditions over land than over sea (higher surface albedo, different aerosol types). In principle, this may lead to systematic different filtering over land than over sea, but since the chosen limits are quite wide we do not expect a large effect. For example, the cloud and albedo correction filter removes only less than 1% of the data.

We will mention this in the paper as follows:

„Because of different conditions over land than over sea (higher surface albedo, different aerosol types), using the same filtering limits may in principle lead to a systematic land-sea bias. However, since the chosen limits are quite wide, this effect is small. For example, the cloud and albedo correction filter removes only less than 1% of the data. "

(Q6) p.9 line 31: Why would higher spatial resolution result in a higher AMF ?

(A6) Higher spatial resolution should not lead to an higher AMF, but the different local time of the measuremenst can play a role. We will clarify this by excluding the term spatial resolution.

(Q7) p.12 line 3: Is there no dependence on viewing angle ?

As mentioned before, viewing geometry differences are handled via the AMF correction. Simulations revealed a small remaining dependence of the AMC-DOAS TCWV on viewing angle, but this e.g. cannot explain the across-track features which are mentioned later. However, the empirical correction handles both effects.

(Q8) p.12, 13. It is unclear from the text how the albedo correction is done in relation to Eq. (1) .
Is it a correction that is only applied to the factor "a" (AMF correction factor) and not to the saturation b_lambda ? Equations (3) and (4) seem to suggest so. But isn't it that not-so-low clouds may have significantly less saturation ? Or is Eq.1 with modified "a" used for the "clear" part of the scene and is there an additional similar equation for the cloudy part with a LUT of a, tau_O2, c_lambda, b_lambda as function of CF, CTH, albedo, SZA, VZA ?
Is it then not a correction at all but rather a calculation method similar to Eq.(1) ?
If it is a correction to Eq. 1, is there any difference to the LUT with fixed surface albedo e.g. in resolution of the parameter grid ? Please clarify and rework the text.

(A8) Equation (1) is the basic AMC-DOAS formula. For the intital retrieval process it is assumed that there is no cloud contamination, i.e. we assume that all TCWV are retrieved for the clear-sky scene only and a fixed surface albedo of 0.02. The correction factor „a" handles in principle all deviations from these assumption. The fitting procedure itself does not include additional cloud and albedo effects, i.e. „b", „c" and „tau" does not depend on CF, CH,albedo, VZA. They depend on the spatial across-track pixel, surface height and SZA. After the nonlinear fit we get the  uncorrected AMC-DOAS S5P TCWV. The albedo/cloud correction takes places afterwards based on additional LUTs which then also depend on albedo and cloud fraction.

We will clarify this in the revised version:

„Note that the albedo-cloud correction is independent from the retrieval and its parameters. It is applied after the retrieval and uses its own look-up tables."

(Q9) p.13 line 11. Please discuss uncertainties in the correction for cloud shielding that are related to the assumption of a fixed H2O profile.

(A9) You are right. There is an additional uncertainty due to the assumption of a fixed H2O profile used for the calculation of correction factor. This will be discussed in the paper as follows:

„For the AMC-DOAS TCWV correction a fixed atmospheric profile is used. The real atmospheric vertical structure of water vapour is highly variable and introduces an uncertainty especially in the AMC-DOAS TCWV cloud correction due to different profile shapes. For example, if there is more water vapour present beneath a cloud than above the cloud layer in comparison to the reference profile this will lead to an underestimation of the actual TCWV."

(Q10) p.13 line 20. There is a very large contrast between ocean albedo and cloud albedo. I would expect that small errors in the FRESCO cloud fraction can induce large errors in surface albedo. Please discuss.

(A10) It is true that the signal received by TROPOMI or other instruments is dominated by the cloud albedo over ocean. If the cloud fraction is too high (low), the retrieved surface albedo is too low (high). This affect also the correction factors, which means that the H2O product quality depends on the accuracy of the cloud product.

We will discuss this in the paper:

„It has to be noted that over ocean where the surface albedo is very low the signal may be dominated by residual cloud. If the cloud fraction is too high (low), the retrieved surface albedo is too low (high). This affects also the correction factors, which means that the AMC-DOAS TCWV product quality depends on the accuracy of the cloud product. "

(Q11) p.14 line 1. Please do not use "instrumental striping" or "striping" for your TCWV dependence on viewing angle. In the OMI / TROPOMI context the word "striping" is generally used to describe a systematic across-track pixel-to-pixel effect caused by systematic differences of the solar calibration spectrum across the CCD. OMI/TROPOMI striping is visible over ocean and over land. Your effect is only over ocean (probably NOT instrumental) and has a very smooth viewing angle dependence. Its origin is definitively different than that what is commonly named "striping".

(A11) Thanks for this comment. We will rename this indeed misleading term to „across track features" in the revised version.

(Q12) p.14 line 2. What do you mean by "our simulation". The (simplified) clear sky model?

(A12) We mean the clear sky model and will clarify this in the text. „Fig. 3d cannot be reproduced by our clear-sky model."

(Q13) p.14 line 2. "Consequently, we assume that they are related to instrumental features"
This conclusion seems very premature to me. Are there no effects that you have neglected in the simulation ??
First of all, why would an instrument effect be visible over ocean but not over land ?
Second, if I look at Fig.2b, it almost looks as if near the sub-satellite point the negative deviation is largest (glitter ?).
In Fig.2d it seems that the albedo correction removes most of that north-south difference over ocean.
Nevertheless, can glitter not be a reason why your simulation doesn't match the observation ?
Same for maritime aerosol which is always present over areas with whitecaps / large wind speeds.
In general, aerosol scattering introduces an azimuth dependence that is not covered by your retrieval model.
Please note that Grossi et al.2015 (Total column water vapour measurements from GOME-2 [...] AMT 8, 1111)  also found in their H2O retrievals a strong scan angle

dependency over ocean, which they attribute to Cox-Munk like BSDF effects.
Your albedo correction may take some of these effects away, but to conclude that a similar effect seen by another retrieval algorithm in another instrument must be instrumental because it doesn't match your theory is a long shot.
There is nothing wrong with an emperical correction if you have a simplified theory that doesn't match observations, but don't blame it on the instrument unless you have a good reason to do so. (also review p.29 line 13)

(A13) The fact that we do not see the across-track features over land does not necessarily mean that the effect is not related to the instrument. For example, straylight may have a larger effect over ocean than over land because of the lower albedo. However, we agree that there are also other potential effects including aerosols, glitter/glint, non-lambertian surface or other limitations of our model. Especially we thank the referee for the reference to the Grossi paper.

We will replace the sentence with the following:

„The reason for this effect is not clear. It could be an instrumental effect due to straylight which may have a larger effect over ocean than over land because of the lower ocean albedo. However, other effects not explicitly included in our model, such as aerosols, glitter/glint and non-lambertian surface relectance may also play a role. In fact, the across-track features are similar to the observation of Grossi et al. (2015) who could attribute this to non-lambertian surface effects."

(Q14) p.14 line 4. "There is no dependence on season or position of the instrument." I think this needs a bit more discussion how you reach that conclusion.Also Grossi et al.2015 fit a polynomial as function of viewing angle, as you do. However, their correction values are not only a function of scattering angle, but also vary with latitude, and they vary from month to month. It may well be that your albedo correction implicitly takes care of these dependencies. That would be interesting to know.And what did you look at to conclude that position of the instrument doesn't play a role? For example, do you get in Fig.4 exactly the same distributions if you split the dataset in northern and southern hemisphere? How is this for December or June when the subsolar point is at maximum/minimum latitude? (or for better statistics: how compare Dec+Jan north/south to Jun+Jul)?
Of course you may still use a single curve for correction if that's you preferred algorithm. But it would be good to know limitations of that approach.

(A14) We mean that we assume that there is no dependence on month. We investigated the temporal behaviour of the empirical correction. The general shape was similar for all months so we decided not to include any temporal dependence. We also do not see a large dependence on solar zenith angle which implies that there is also no latitude and seasonal dependence. Note that varying viewing geometry is already handled by the AMF correction, so this should already remove many dependencies. The albedo correction handles further parts of the across-track features (e.g. sun-glint is removed).

We will replace this sentence by:

„We investigated the temporal behaviour of the empirical correction. The general shape was similar for all months so we decided not to include any temporal dependence. We also

do not see a large dependence on solar zenith angle. Note that varying viewing geometry is already accounted for by the air mass factor correction, so this already removes many dependencies. The albedo correction handles further parts of the across-track features (e.g. sun-glint is removed). "

(Q15) p.17 line 19. "The averages are calculated with weighting according to the latitude." Please be a bit more specific how the weighting is done.

(A15) The weighted average is done according to cos(latitude). This will be clarified.

(Q16) Section 4.1.3 and 4.1.4 Do MPIC and SRON use the same surface height as AMC-S5P ?

(A16) MPIC uses as source for the surface height the values written to the NO2 product files which are based on the GMTED2010 surface elevation database. This is also used for the AMC-DOAS products. The SRON height information is written to the CO product files which is also based on GMTED2010. Difference arises from the lower spatial resolution of the instrument in the shortwave infrared bands. This should only be relevant in regions with large surface height variations like mountain areas. In fact, our comparisons of the independent products does not show this.

(Q17) p.22 line 9. What is the difference between FRESCO used by AMC-S5P and FRESCO-S used by MPIC-S5P ?

(A17) For S5P AMC-DOAS, FRESCO is the support product for TROPOMI provided by KMNI. Borger et. al (2020) use the cloud information from the TROPOMI L2 NO2 product which is also based on the FRESCO algorithm, but is optimized to retrieve NO2. Especially, the spatial resolution of both products is different. Our FRESCO has the same spatial resolution as our retrieval window and is already collocated.

We will include the following explanation within the text:

„FRESCO-S is a specific cloud product based on the FRESCO algorithm, but adapted to the retrieval of $NO_2$. It differs from the FRESCO cloud product used for the AMC-DOAS, for example in the spatial resolution."

(Q18) p.24 Fig.8 Can you show the average cloud cover in this month (as proxy the number of filtered/valid observations) ? Is there a relation to negative values (e.g. in the tropical rainforsest)?

[Figure]

(A18) The image above shows the monthly sum of the daily counted valid AMC-DOAS S5P TCWV data within one gridbox (left column). The right column shows the deviation between the

ERA5 TCWV and AMC-DOAS S5P TCWV. The counts can be used as a rough estimate of the cloud fraction filter within the tropics. Actually the negative values indeed correlate with the regions with low data availability, e.g in the tropical rain forest. This means that probably many cloudy scenes have been filtered out. However the ERA5 data are filtered in the same way before the averaging (only collocated data are used). Therefore this should not be a sampling issue. However, this indicates that the observed differences at these regions are probably less reliable. We will mention this in the text as follows:

„The observed negative differences between ERA5 TCWV and AMC-DOAS S5P TCWV at tropical regions, e.g. tropical rain forests, correlate with regions of low data availability. We infer that these values are possibly less reliable."

(Q19) p.28 line 17. The "tongues" in southern parts of the pacific have also turned up in various other comparisons between optical sensors and SSMI / SSMIS. It may be associated to cloud cover, therefore see also previous question.

(A19) We do not see a clear relation to the number of valid data in this case.

(Q20) p.29 line 22. For clarity, add "global average" or say this is the systematic global bias.

(A20) We will add this.

(Q21) p.29 line 30. "The standard deviation of the differences..." Which differences ? Global monthly means ? Monthly means per grid-cell ? Daily means per grid cell (based on 2 months) ? You need to be more specific here. This comment also applies to similar statements in the Abstract.

(A21) We will clarify the use of differences in the text.

(Q22) p.30 line 4. The natural variability should be smaller than this, otherwise the measurements / retrievals would be error-free which they are not. Apart from the fact that "natural variability" is a dangerous term because it varies with spatial / temporal differences. Please be more precise and reword.

(A22) We agree and we will remove this sentence on natural variability because it is actually not necessary.

We thank you for your constructive feedbacks and comments. We will address them in the revised paper.